# MVDGC: Joint 3D and 2D Multi-view Pedestrian Detection via Dual Geometric Constraints

**Thinh Phan**                                                      *thinhp@uark.edu*
*AICV Lab, EECS Department, University of Arkansas*

**Hao Vo**                                                          *haov@uark.edu*
*AICV Lab, EECS Department, University of Arkansas*

**Khoa Vo**                                          *khoavoho@uark.edu@uark.edu*
*AICV Lab, EECS Department, University of Arkansas*

**Thanh Ngo**                                          *ngoduc.thanh@uit.edu.vn*
*University of Information Technology*

**Cuong Pham**                                              *cuongpv@ptit.edu.vn*
*Posts and Telecommunications Institute of Technology (PTIT), Vietnam*

**Ngan Le**                                                         *thile@uark.edu*
*AICV Lab, EECS Department, University of Arkansas*

**Reviewed on OpenReview:** *https://openreview.net/forum?id=4OcVQX5Mxc*

## Abstract

The core challenge in multi-view pedestrian detection (MVPD) lies in effective aggregation of visual features from different viewpoints for robust occlusion reasoning. Recent approaches have addressed this by first projecting image-view features onto a Bird's Eye View (BEV) map, where ground localization is then performed. Despite impressive performance, the perspective transformation induces severe distortion, causing spatial structure break and degrading the quality of object feature extraction. The blurred and ambiguous features hinder accurate BEV point localization, especially in densely populated regions. Moreover, the strong mutual relationship between the BEV ground point and image bounding boxes is not capitalized on. Although multi-view consistency of 2D detections can serve as a powerful constraint in BEV space, these detections are commonly treated as auxiliary signals rather than being jointly optimized with the primary task.

In this work, we propose **MVDGC**, a unified framework that *jointly estimates pedestrian locations on the BEV plane and 2D bounding boxes in image views*. MVDGC employs a *sparse set of 3D cylindrical queries* that embraces geometric context across both BEV and image views, enforcing dual spatial constraints for precise localization. Specifically, the geometric constraints is established by modeling each pedestrian as a vertical cylinder whose center lies on the BEV plane and whose projection casts a rectangular box in the image views. These queries function as shape anchors that directly extract 2D features from the intact image-view features using camera projection, eliminating projection-induced distortions. The 3D cylindrical query enables the unification of BEV and ImV localization into a single task: 3D cylinder position and shape refinement.

Extensive experiments and ablation studies demonstrate that MVDGC achieves state-of-the-art performance across multiple evaluation metrics on MVPD benchmarks, including WildTrack and MultiViewX. On the generalized multi-view detection (GMVD) dataset, MVDGC achieves the highest MODP and precision, while maintaining competitive perfor-

mance on the remaining metrics, highlighting its robustness and generalization to unseen scene configurations. Code is available at: `https://github.com/UARK-AICV/MVDGC`

## 1 Introduction

Occlusion remains a major challenge in monocular multi-object detection, particularly in crowded scenes. A common solution is to employ multi-camera systems with overlapping fields of view, where occluded objects in one view can be recovered through complementary perspectives from other cameras. This setting motivates multi-view object detection, which plays a key role in applications like surveillance Zhang et al. (2021b); Baqué et al. (2017); Zhang & Chan (2019), autonomous driving Liu et al. (2023); Li et al. (2024); Wang et al. (2023b), and 3D pose estimation Zhang et al. (2021a); Choudhury et al. (2023); Wang et al. (2023a). The key challenge is how to effectively aggregate information across views. In this study, we focus on multi-view pedestrian detection (MVPD) with the aim to accurately localize pedestrians on a shared ground plane under dense crowds and severe occlusion.

Although MVPD and 3D object detection operate in analogous multi-camera scene settings, their problem formulations and supervision regimes differ substantially. In 3D object detection, the datasets Caesar et al. (2020); Chang et al. (2019); Sun et al. (2020) provide full 3D bounding box annotations that specify an object's position, size, and orientation ($x$, $y$, $w$, $h$, $l$, $roll$, $pitch$, $yaw$) in the world coordinate system. Such rich supervision enables query-based frameworks to explicitly reason about object geometry and to aggregate features from multiple views through well-defined 3D projections. Properties of 3D object detection approach is summarized in Table 1 (first column). In contrast, MVPD datasets focus on localization on the ground plane and image plane, consisting of pedestrian ground-plane locations ($x_{BEV}$, $y_{BEV}$) and 2D bounding boxes ($x_{img}$, $y_{img}$, $w_{img}$, $h_{img}$) in individual camera views. Given this limited annotations, existing MVPD methods, generally fall into two categories: anchor-based and centralized-based approaches, as illustrated in Fig. 1(a and b). Anchor-based methods (e.g., Deep-Occlusion Baqué et al. (2017), DeepMCD Chavdarova & Fleuret (2017), GNN-CCA Luna et al. (2022)) first perform 2D detection independently in each camera view and then associate detections across views using geometric proximity on the ground plane and appearance similarity. While computationally efficient, these methods are vulnerable to occlusion, missed detections, and weak cross-view association. Centralized-based methods (e.g., MVDet Hou & Zheng (2021), 3DROM Qiu et al. (2022), EarlyBird Teepe et al. (2024a)) address these limitations by projecting image features into a common BEV representation and performing dense occupancy-based detection. However, perspective projection introduces severe geometric distortion, stretching features non-uniformly with respect to camera-object distance. Even with enhanced feature-pulling strategies Lee et al. (2023); Aung et al. (2024), the reliance on discretized BEV heatmaps leads to quantization errors that ultimately limit localization precision. Properties of anchor-based and centralized-based methods are summarized in Table 1 (second and third columns, respectively). A key question arises: *"Can we bypass the projection distortion and eliminate the dependency on centralized representation, while still achieving robust multi-view detection?"*

To answer this, we approach ground-plane (BEV) localization from the perspective of ImV reasoning. Assuming that each pedestrian could be represented as a uniform 3D geometric shape, accurate localization can be achieved when two conditions are satisfied: (i) the projected regions of the 3D object closely align with the 2D object regions in each visible view, and (ii) the visual features extracted from these regions remain highly consistent across views, reflecting the strong cross-view appearance correlation of individual pedestrians. In another term, the stronger we impose the geometric and appearance constraints on the 2D projections, the sharper and more precise the 3D detection becomes. However, centralized-based methods often overlook this alignment, treating 2D detection as auxiliary, despite the fact that the image foot point and BEV ground point refer to the same real-world location. Without an explicit bidirectional mapping between BEV ground points and ImV bounding boxes, they fail to exploit this correspondence. On top of that, the explicit bridge between BEV ground points and ImV detections can be leveraged to support robust tracking in dense 2D scenes, as the globally consistent BEV representation provides stable identity cues across views and time. By associating detections through a shared 3D geometry, object identities can be propagated across cameras and maintained despite occlusion, viewpoint changes, and intermittent detection failures in individual views. While representing object as 3D bounding box has been widely adopted in 3D object detection Li et al.

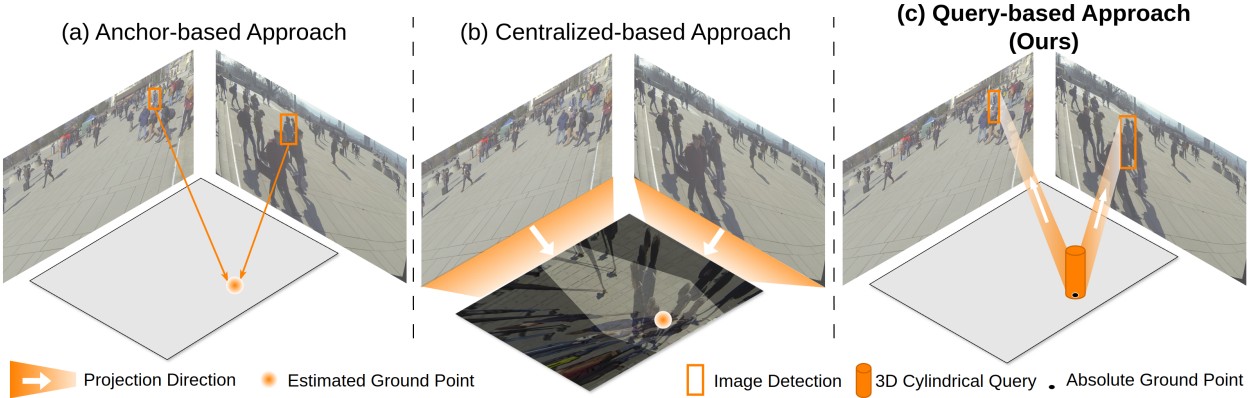

Figure 1: High-level comparison between anchor-based methods (a), centralized-based methods (b), and our proposed query-based approach (c). (a) estimates locations by associating image detections via proximity in ground plane projection and appearance similarity. (b) aggregates perspective-transformed image features into a shared BEV space, localizing by predicting dense occupancy map. (c) We propose a query-based detection framework that refines each 3D pedestrian cylindrical query in 3D space by aggregating and aligning multi-view image features projected from the query itself.

Table 1: Comparison among 3D Box Query, Anchor-based, Centralized-based, and our proposed MVDGC.

|  | **3D Box Query** (DETR3D, BEVFormer) | **Anchor-based** (DeepMCD, GNN-CCA) | **Centralized-based** (MVDet, EarlyBird) | **3D Cylindrical Query** (**Proposed MVDGC**) |
|---|---|---|---|---|
| **Object Representation** | Sparse 3D bounding box | Dense BEV grid | Dense BEV grid | Sparse 3D cylinder |
| **Input Features** | Feature sampling from raw multi-view features | Appearance features & 2D bounding boxes | Projected feature maps (distorted) | Feature sampling from raw multi-view features |
| **Image Projection** | 2D non-uniform polygon | 2D point | 2D point | 2D rectangular shape |
| **Provided Annotation** | Full 3D bounding boxes (x,y,z,w,h,roll,pitch,yaw) | Ground points (x,y) & 2D bounding boxes | Ground points (x,y) & 2D bounding boxes | Ground points (x,y) & 2D bounding boxes |
| **BEV Supervision** | 3D box regression | BEV regression | BEV regression | Dual BEV–ImV supervision |
| **3D Shape Supervision** | Fully | ✗ | ✗ | Weakly (Radius/height inferred from 2D boxes) |
| **3D–2D Consistency** | ✗ | ✗ | ✗ | ✓ (Achieved via BEV↔ImV bridge) |
| **Handle Distortion** | ✓ | ✗ | ✗ | ✓ (Eliminated via direct multi-view sampling) |

(2024); Wang et al. (2022; 2023b), the lack of full 3D annotation in multi-view pedestrian datasets makes the regression of 3D box infeasible. To address this, we formulate each 3D object as a vertical cylinder regarding the human walking posture. As illustrated in Figure 1 (c), the cylinder, which is anchored at the ground point along with its projections forming the 2D bounding boxes tightly enclosing a pedestrian, conveniently encodes the geometric information shared across both BEV and ImV domains. Compared to a 3D rectangular cuboid, whose projection results in a polygon with more than four vertices, the projection of a cylinder produces a clean rectangular box in the image plane. Hence, the projection of the cylinder as ImV detection could directly support the BEV localization, and weakly supervise the shape $(w, h)$ of 3D object. Instead of separately regressing BEV and ImV outputs, our formulation unifies both tasks by solely refining the cylinder's absolute position and shape, enabling joint supervision across both domains. To highlight the our novelty, Table 1 demonstrates the detailed comparison between existing MVPD methods, 3D object detection methods, and our novel object formation.

Based on these observations and studies, we propose **MVDGC**, an end-to-end query-based framework for multi-view pedestrian detection that jointly predicts both ground positions and ImV bounding boxes without relying on dense BEV feature maps. The pipeline of our proposed method is illustrated in Figure 2. Specifically, in the first step, MVDGC introduces a set of learnable 3D cylindrical queries distributed over the ground plane. The second step is conducted by the three modules, namely *Multi-view Feature Sampling*, *Intra-Inter Query Interaction*, and *Multi-view Adaptive Fusion*. The *Multi-view Feature Sampling* module

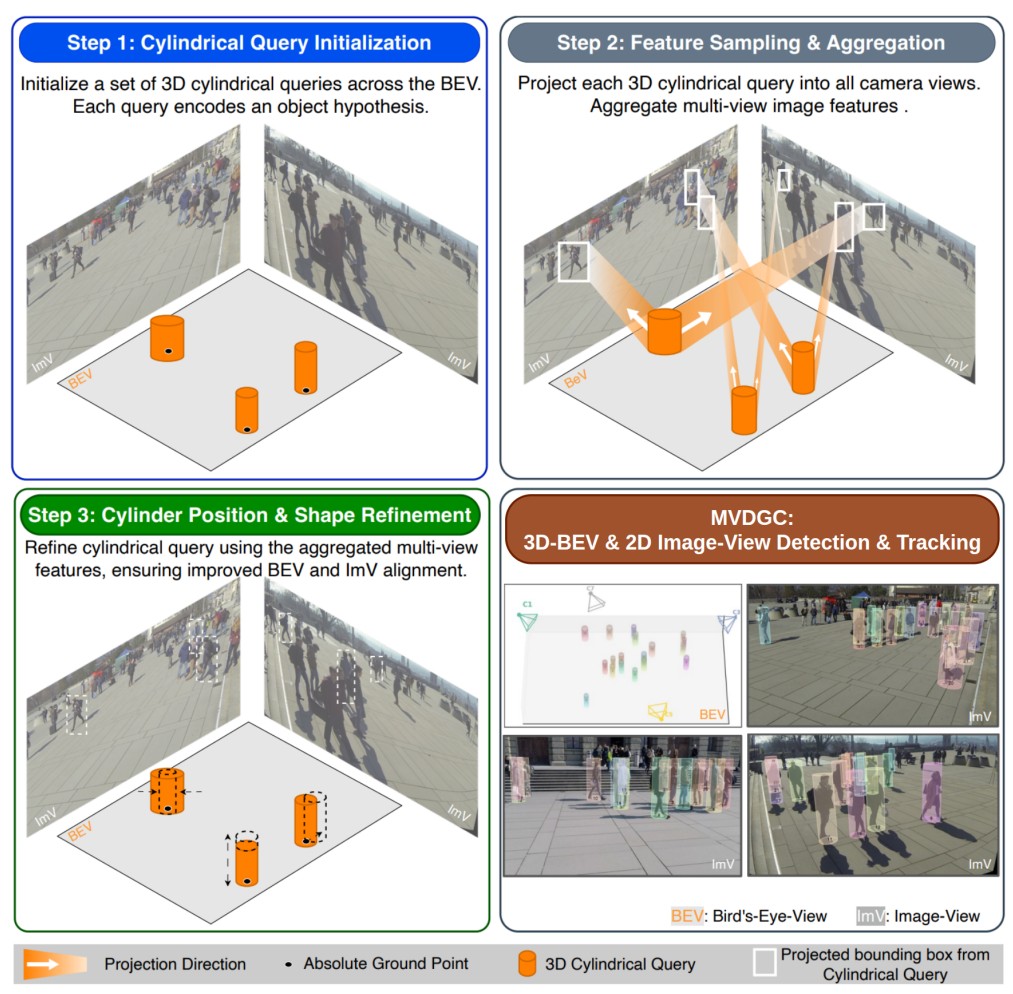

Figure 2: Pipeline of the proposed MVDGC framework. Out method represents each pedestrian as a 3D cylindrical query defined in Bird's-Eye View (BEV) space in step 1. These queries are projected into all camera views to sample and aggregate multi-view image features in step 2. As shown in step 3, the combined features are then used to iteratively refine the cylinders' spatial positions and shapes by geometric constraints on both BEV and image-view (ImV) domains. The shared representation as the vertical cylinder enables joint 3D-BEV localization, and 2D ImV detection and tracking.

projects each cylindrical query into the image planes through camera parameters and gathers visual features from raw image features. These coarse features are then passed to the *Intra-Inter Query Interaction* module, which facilitates the information exchange among queries in the same view (Intra) as well as queries of the same objects across views (Inter). The aggregated multi-view information is adaptively fused in the *Multi-view Adaptive Fusion* module. In the final step, the aggregrated features are used to refine each cylinder's spatial attributes. The predicted BEV point and ImV bounding boxes, derived from the refined cylinder, are supervised by minimizing BEV localization error and maximizing overlap with ImV ground truth boxes. In contrast to existing anchor-based and centralized-based MVPD approaches, MVDGC avoids dense BEV representations and perspective projection, thereby eliminating quantization errors and projection-induced distortions. By explicitly bridging BEV ground points and ImV detections through a unified 3D cylindrical query, MVDGC establishes a consistent global identity across BEV and ImV domains, enabling joint detection and robust tracking under severe occlusion. This object-centric formulation allows MVDGC to more effectively exploit multi-view geometric consistency while remaining compatible with the limited annotations available in MVPD datasets.

Our main contributions are summarized as follows:

- We propose **MVDGC**, a novel framework for MVPD that directly regresses BEV coordinates, thereby avoiding quantization errors and projection distortions introduced by BEV feature maps. MVDGC is built upon **three key components**, named as *Multi-view Feature Sampling*, *Intra-Inter Query Interaction*, and *Multi-view Adaptive Fusion*, which together enable effective multi-view feature aggregation and robust object-centric reasoning.

- We introduce the **3D cylindrical representation** that unifies BEV localization and ImV detection under coherent geometric constraints. This dual constraints across domains result in the highest localization precision on the ground plane on multiple benchmarks. Moreover, the cylindrical representation establishes a shared global identity between BEV and ImVs, enabling joint detection across spatial domains and robust single-view tracking under severe occlusion.

- We conduct **extensive experiments** on the WildTrack, MultiViewX, and GMVD datasets, demonstrating superior performance on pedestrian localization. Additionally, we validate the BEV-ImV integrity of our unified representation by transferring the BEV tracking results to a downstream single image-view tracking task, which outperforms other single-view tracking methods.

## 2 Related Work

### 2.1 Anchor-based Approaches

Early multi-view pedestrian detection methods typically relied on 2D detection results to infer the 3D spatial relationships via probabilistic occupancy modeling. Fleuret et al. (2008) estimated the probability of pedestrian presence on the ground plane by applying a generative model to background-subtracted images. Building on this idea, Roig et al. (2011) proposed a framework based on Conditional Random Fields (CRFs), where each camera view had its own set of nodes and occlusion handling. Baqué et al. (2017) extended the CRF formulation by incorporating high-order terms, measuring discrepancies between convolutional neural network (CNN)-based body part predictions and a generative occlusion-aware model. Chavdarova & Fleuret (2017) first trained a monocular occlusion-aware pedestrian detector, then fused detections across views to enhance appearance modeling and multi-view reasoning. However, these anchor-based methods lack early feature alignment, limiting joint reasoning under occlusion or partial visibility. *Unlike anchor-based approaches, our MVDGC performs object-centric feature aggregation across views at an early stage through a shared 3D query, enabling coherent multi-view reasoning and robust localization even under severe occlusion.*

### 2.2 Centralized-based Approaches

Recent research has shifted toward localization on centralized plane, in which image features from multiple views are projected into unified representation, typically Bird's Eye View (BEV) plane. MVDetHou et al. (2020) pioneered this direction by projecting image-view (ImV) features onto the ground plane and built the pedestrian occupancy maps through spatial aggregation. This method was later enhanced by MVDetr Hou & Zheng (2021) by employing the deformable attention mechanism to push the shadow-like features to their correct locations and adaptively aggregate features across views and positions. SHOT Song et al. (2021) introduced multiple homographies to project features at different heights to counter ambiguities in object appearances. 3DROM Qiu et al. (2022) tackled robustness by introducing a 3D random occlusion mechanism during training. Despite improved performance over anchor-based methods, centralized-based approaches still face challenges due to distortion induced by perspective projections. To address this issue, MVTT Lee et al. (2023) first extracted features from 2D detection regions of interest and projected them onto estimated foot positions on the ground plane. MVFP Aung et al. (2024) employed a non-parametric 3D feature pulling mechanism that directly extracts 2D image features for each valid voxel in the 3D BEV volume, effectively mitigating feature distortion and improving BEV detection quality. PVH Alturki et al. (2025) leveraged visual hull to complement 3D feature pulling with pedestrian potential locations. *Unlike centralized-based approaches, our MVDGC completely avoids constructing dense BEV feature maps and perspective projection altogether. Instead, MVDGC adopts an object-centric, query-based formulation that directly samples features from raw ImVs using a unified 3D cylindrical representation, thereby eliminating projection-induced distortion and quantization errors while enabling consistent multi-view reasoning across BEV and ImV domains.*

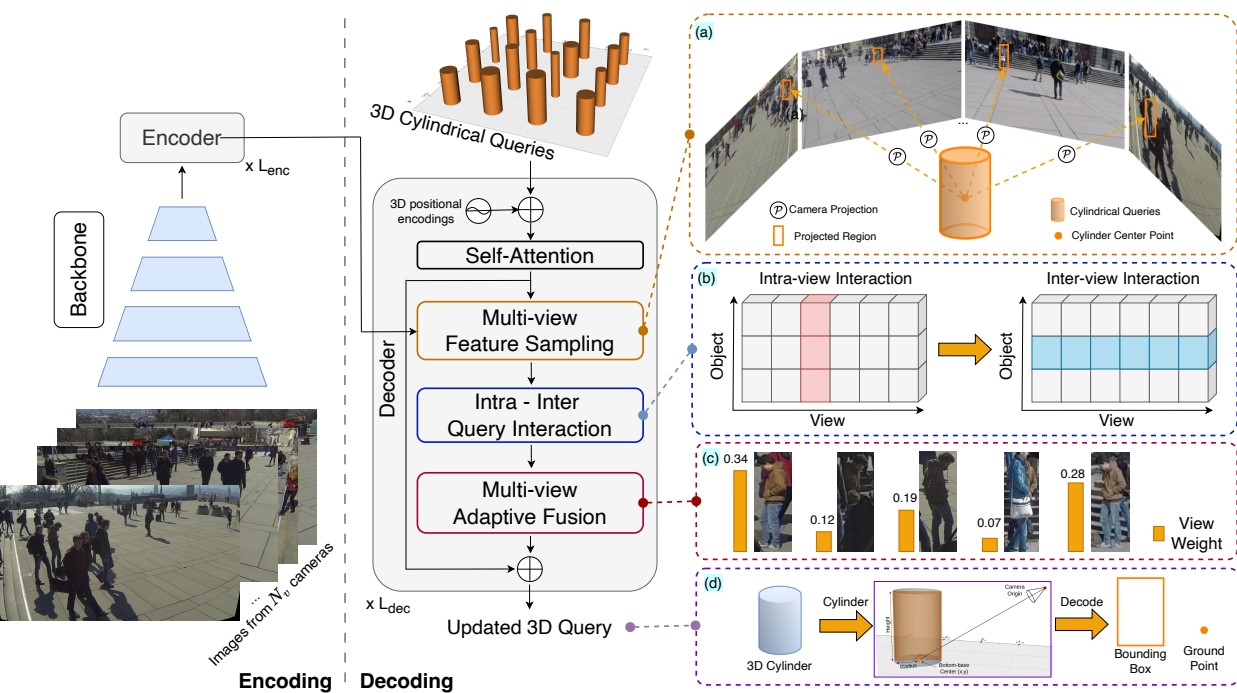

Figure 3: The overall architecture of MVDGC. At the encoding stage, multi-view features are extracted by an image backbone and then enhanced by the encoder. At the decoding stage, 3D cylindrical queries are fed to the decoder which includes the following modules. The *Multi-view Feature Sampling* (a) module projects 3D queries into 2D queries across views and navigates the multi-view feature extraction. Then, the *Intra-Inter Query Interaction* (b) module strengthens 2D query representations through in- and cross-view exchange. The *Multi-view Adaptive Fusion* (c) module weighted consolidates multi-view features, which are then used to refine 3D cylindrical queries with dual constraints (d) from BEV and ImVs.

*These related work collectively highlight two insights: (1) preserving foreground feature integrity prior to fusion is critical, and (2) excessive reliance on 3D BEV with centralized representation and overlooking ImV supervision can limit performance. In contrast to the aforementioned methods, our approach MVPD eliminates BEV projection distortion by adopting a query-based detection framework. Our MVDGC uses learnable 3D cylindrical queries that interact directly with ImV features, enabling accurate spatial reasoning without relying on intermediate representations. Importantly, we enforce strong BEV-ImV consistency by supervising predictions with both ground plane and image views, leading to robust multi-view detection.*

## 3 Methodology

### 3.1 Overview

The overall architecture of MVDGC framework is illustrated in Figure 3 with two stages as follows:

**Encoding:** At the encoding stage, MVDGC takes a set of synchronized RGB images $\mathcal{I} = \{I_v\}_{v=1}^{N_v}$ from $N_v$ camera views as input, where $I_v \in R^{3 \times H \times W}$ represents a single-view color image of height $H$ and width $W$ captured by the camera $v^{th}$. A frozen pre-trained image backbone $\phi$ extracts multi-scale features $\mathcal{S} = \{S_v\}_{v=1}^{N_v}$ from $\mathcal{I}$, where $S_v = \phi(I_v)$. The multi-scale features $\mathcal{S}$ is then further encoded by a Transformer encoder Carion et al. (2020) with $L_{enc}$ layers, enabling flexible adaptation of the features to our task.

**Decoding:** At the decoding stage with $L_{dec}$ decoder layers, we initialize a sparse set of 3D cylindrical queries on the BEV plane to serve as anchors (Section 3.2). These 3D queries first undergo self-attention, enabling early spatial reasoning by facilitating contextual interaction among queries. The *Multi-view Feature Sampling* (Section 3.3) module then decomposes and projects them into each visible ImV using the camera projection

matrices, forming 2D queries. These 2D queries are used to sample relevant image features across multiple views. To strengthen 2D query representations, the *Intra-Inter Query Interaction* (Section 3.4) module facilitates query information exchange along individual views and across multiple views. The aggregated multi-view features are adaptively fused by *Multi-View Adaptive Fusion* (Section 3.5) to enrich the 3D queries. Finally, the updated 3D queries are not only supervised by BEV ground points, but also weakly constrained by enforcing the overlap between their image projections and the corresponding 2D bounding boxes (Section 3.6).

## 3.2 Cylindrical Query Formulation and Initialization

To enable simultaneous BEV and ImV outputs, we represent each query $Q^{3D} = (F^{3D}, P^{3D})$ in BEV space as a learnable 3D cylindrical anchor, composed of position component $P^{3D} \in \mathbb{R}^{N_q \times 4}$ and feature component $F^{3D} \in \mathbb{R}^{N_q \times D}$ where $N_q$ is the number of queries and $D$ is feature dimension. The decouple of position and feature components provides better positional prior, accelerating convergence and improving localization performance. While the feature component $F^{3D}$ stores the appearance information, the position component $P^{3D}$ characterizes the 3D cylindrical query with four variables $P^{3D} = \{x, y, h, r\}$, where: ground-plane coordinates $(x, y)$ and shape parameters $(r, h)$, where $r$ is radius and $h$ is height. Initially, $N_q$ vertical cylindrical queries are distributed evenly over a grid covering the surveillance area. During training, $F^{3D}$ is updated through multi-view feature aggregation managed by the three subsequent modules in each decoder layer, and then used to predict the alignment and refine $P^{3D}$. This process is repeated across $L_{\text{dec}}$ decoder layers which progressively regress the final $P^{3D}$.

## 3.3 Multi-view Feature Sampling

We first inject 3D positional encodings into $F^{3D}$ via element-wise addition and apply self-attention, allowing the feature component to capture spatial and semantic relationships within the set. To enable $Q^{3D}$ to sample the ImV feature, we project $F^{3D}$ to per ImV representations $F^{2D} = \{f_v^{2D}\}_{v=1}^{N_v} \in \mathbb{R}^{N_v \times N_q \times D}$ through a linear layer for all cameras. $F^{2D}$ are then paired with their corresponding 2D reference points $P^{2D} = \{\mathbf{u}_v^{2D}\}_{v=1}^{N_v} \in \mathbb{R}^{N_v \times N_q \times 2}$. 2D reference points $\mathbf{u}_v^{2D}$ corresponding to image $I_v$ are obtained by projecting the middle-center points $\mathbf{u}^{3D} = (x, y, h/2)$ of $P^{3D}$ with projection matrix $\mathcal{P}_v \in \mathbb{R}^{3 \times 4}$ of camera $v^{th}$. As illustrated in Figure 3(a), for each camera view $v^{th}$, we apply the following formulation to $N_q$ anchors:

$$[\mathbf{u}_v^{2D}; 1] = \mathcal{P}_v[\mathbf{u}^{3D}; 1] \tag{1}$$

Following the deformable cross-attention mechanism Zhu et al. (2020), we utilize $F^{2D}$ as query to sample features around the projected points $\mathbf{u}^{2D}$ in image feature maps $\mathcal{S}$. Instead of warping or resampling the entire image feature map into a BEV grid, this operation selectively gathers local features directly in the image domain, where spatial relationships are preserved. As a result, the model avoids the geometric distortion and resolution loss commonly introduced by perspective projection, while still leveraging accurate multi-view geometric correspondence through camera projection.

## 3.4 Intra-Inter Query Interaction

The obtained features $F^{2D}$ of 2D queries from previous module decently carry the appearance information. However, they are coarse, exclusive, and dissociated in 2D space. Therefore, it is crucial to model the contextual relationship among queries in each view and of the same query across views of the same object. However, simply applying self-attention over all dimensions is very costly but do not well-reflect the semantic and geometric relationships, which is already studied in space-time problem in video domain Bertasius et al. (2021). Following this reasoning, we introduce the Intra-Inter Query Interaction module, as depicted in Figure 3(b), which factorizes the interactions into two steps: intra-view and inter-view, applied sequentially. The intra-view interaction allows for communication of object features in the same view, whereas the inter-view interaction reinforces the homogeneity of appearance features from the same object across views. We formulate both interactions via *masked self-attention operators*. First, $F^{2D}$ is flattened into $\tilde{F}^{2D} \in \mathbb{R}^{(N_v \cdot N_q) \times D}$. Given an index $i \in \{0, ..., N_v \cdot N_q - 1\}$ corresponding to the $i \bmod N_q$-th query and the $\left\lfloor \frac{i}{N_q} \right\rfloor$-th view, the masks

$M$ are constructed with flatten indices as follows:

$$M_{\text{intra}}[i,j] = \begin{cases} 1 & \text{if } \left\lfloor \frac{i}{N_q} \right\rfloor = \left\lfloor \frac{j}{N_q} \right\rfloor \\ 0 & \text{otherwise} \end{cases}$$

$$M_{\text{inter}}[i,j] = \begin{cases} 1 & \text{if } (i \bmod N_q) = (j \bmod N_q) \\ 0 & \text{otherwise} \end{cases}$$

$$(2)$$

Both masks are of shape $\mathbb{R}^{(N_v \cdot N_q) \times (N_v \cdot N_q)}$, where $M_{\text{in}}$ enables attention among 2D features from same view, and $M_{\text{cross}}$ enables attention among 2D features corresponding to the same 3D cylindrical query. In these masks, entries with value 1 permit attention between query pairs, while entries with 0 blocks the attention. Two-step masked self-attention is applied as follows:

$$\text{Intra-Interaction: } \tilde{F}^{2D}_{in} = \texttt{Self-Attention}(\tilde{F}^{2D}, M_{in})$$

$$\text{Inter-Interaction: } \tilde{F}^{2D}_{cross} = \texttt{Self-Attention}(\tilde{F}^{2D}_{in}, M_{cross})$$

$$(3)$$

Finally, the result $\tilde{F}^{2D}_{cross}$ is re-shaped back to $\mathbb{R}^{N_v \times N_q \times D}$, forming multi-view features $F'^{2D}$ as the outcomes. The ordering of the intra-inter interaction is inspired by video understanding, where modeling spatial dependencies precedes temporal reasoning to enable effective multi-frame fusion. This design allows each query to first establish spatial coherence within individual views by learning local geometric and appearance consistency, before reasoning about cross-view correspondences across cameras.

### 3.5 Multi-view Adaptive Fusion

At this stage, the multi-view features $F'^{2D}$ from $N_v$ cameras must be fused, assisting the update of the cynlindrical feature $F^{3D}$. As depicted in Figure 3(c), due to out-of-sight projection, and full or partial occlusions in some views, ImV appearance features may not equally contribute to the refinement of 3D object representations $F^{3D}$. Although multi-view aggregation has been explored in prior work, existing solutions remain limited in their ability to model such object-dependent visibility. In particular, centralized-based methods Zhang et al. (2024); Hou et al. (2024) typically perform view-level weighting by assigning a single scalar weight to each camera, implicitly assuming uniform contribution across all objects within the same view. This assumption overlooks the fact that different pedestrians may exhibit drastically different visibility and occlusion patterns even under the same camera. To address this limitation, we introduce an adaptive feature-pooling mechanism based on weighted summation, which selectively emphasizes view-specific information according to its relevance and reliability for each object. Enabled by our query-based, object-centric framework, the model learns instance-specific attention weights for every object query in every camera view. This fine-grained fusion strategy allows the network to prioritize the most informative views for each individual pedestrian, rather than applying a uniform view weight across all instances.

Specifically, we estimate a per-query weight by applying a linear transformation followed by softmax normalization on across views of each query:

$$W = \texttt{softmax}(\texttt{Linear}(F'^{2D})) \in \mathbb{R}^{N_v \times N_q}$$

$$(4)$$

These weights are then used to compute a weighted sum of the view-specific features, resulting in an aggregated multi-view feature representation $F^{MV} \in \mathbb{R}^{N_q \times D}$:

$$F^{MV} = \Sigma_{v=1}^{N_v} F'^{2D}_v \odot W_v$$

$$(5)$$

where $W_v \in \mathbb{R}^{N_q \times 1}$ denotes the attention weights for the $v^{\text{th}}$ view of $W$ and broadcasted along the feature dimension of $F'^{2D}_v$, shown as orange bars ▮, where bar's height indicate the weight value in Figure 3(c). The refined 3D features $\hat{F}^{3D} \in \mathbb{R}^{N_q \times D}$ are obtained by merging fused features $F^{MV}$ with initial 3D features $F^{3D}$:

$$\hat{F}^{3D} = F^{3D} + LayerNorm(F^{MV})$$

$$(6)$$

Finally, $\hat{F}^{3D}$ is passed through a Multi-layer Perceptron (MLP) to predict spatial offsets $\Delta P^{3D}$, which is used to refine the postion component $P^{3D}$. The process of postion update can be defined as:

$$\Delta P^{3D} = MLP(\hat{F}^{3D}), \tag{7}$$

$$\hat{P}^{3D} = P^{3D} + \Delta P^{3D} \tag{8}$$

### 3.6 Cylinder Decoding

As shown in Figure 3d (d), the predicted cylinder $\hat{P}^{3D}$ is supervised by point regression in BEV and bounding box detection in ImV. To enforce this dual-constraint, we first decode the cylinder into a BEV point on the ground plane and an ImV bounding box. The BEV point is obtained directly from ground-plane coordinates $(x, y)$ of $\hat{P}^{3D}$. To acquire the ImV bounding box, we need to find the four corners of the projected cylinder on ImV, which can be done efficiently by utilizing camera pose Hartley & Zisserman (2003). Primarily, we determine the diameter of the cylinder's bottom base that is orthogonal to the viewing direction from the camera. Specifically, let the viewing direction be defined by the line connecting the camera center and the cylinder's bottom-base center. We then compute the line passing through the bottom-base center that is perpendicular to this viewing direction, and use its intersection with the circular base to obtain the corresponding diameter. The intersection points between this diameter and the bottom circular edge are the bottom left and bottom right corners of the bounding box in 3D. The top left and top right corners are easily calculated by adding the cylinder's height to bottom corners. Afterall, we project these four corner points to corresponding ImV via camera projection matrix and obtain the bounding box. Thanks to the cylinder representation, we are able to exploit the constraints on dual spaces (cylinder's center as BEV points and its image projection as ImV bounding box). Additionally, the unified representation brings global identity across BEV and ImV, facilitating the downstream task such as ImV tracking, as shown in Table 4 and Table 6.

### 3.7 Training Objectives

Our MVDGC framework optimizes the 3D cylindrical queries on both BEV and ImV spaces, thus its objective functions are conducted as follows:

**BEV Learning.** Unlike centralized-based approaches, our model directly regresses the absolute attributes of each cylindrical query on the ground plane. Since multi-view pedestrian datasets often suffer from annotation ambiguities (e.g., misaligned ground points and inconsistent annotations across views), we adopt a KL-based regression loss He et al. (2019) to explicitly model uncertainty:

$$L_{KL}^{bev} = e^{-\alpha} \left( \left| \tilde{P}^{3D} - \hat{P}^{3D} \right| - \frac{1}{2} \right) + \frac{1}{2}\alpha, \quad \alpha = \log(\sigma^2), \tag{9}$$

where $\tilde{P}^{3D}$ and $\hat{P}^{3D}$ denote the ground-truth and predicted BEV locations, respectively, and $\sigma$ represents the predicted variance that captures sample-wise uncertainty. By treating BEV regression as a distribution-matching problem, the KL loss effectively alleviates label noise and stabilizes optimization under imperfect annotations.

In addition to regression, we apply a focal-style classification loss to distinguish pedestrians from background on the BEV domain:

$$L_{cls}^{bev} = -\alpha_t (1 - p_f)^{\gamma} \log(p_f), \tag{10}$$

where $p_f$ denotes the predicted foreground probability, $\alpha_t$ is the class-balancing factor, and $\gamma$ is focusing parameter.

**ImV Learning.** For ImV supervision, we impose bounding box constraints on the projections of the cylindrical queries. Specifically, we apply a Smooth L1 loss to the bounding box center and scale:

$$L_{smooth\_L1}^{img} = \begin{cases} 0.5(\hat{y}_i - y_i)^2, & \text{if } |\hat{y}_i - y_i| < 1, \\ |\hat{y}_i - y_i| - 0.5, & \text{otherwise,} \end{cases} \tag{11}$$

where $\hat{y}_i$ and $y_i$ denotes the predicted and groundtruth bounding boxes. We also apply a Generalized IoU (GIoU) loss on the bounding box corners:

$$L_{GIoU}^{img} = 1 - \left( \frac{|A \cap B|}{|A \cup B|} - \frac{|C \setminus (A \cup B)|}{|C|} \right),  \tag{12}$$

where $A$ and $B$ denote the predicted and ground-truth bounding boxes, respectively, and $C$ is the smallest enclosing box covering both.

Since annotations for the cylinder radius and height are not available, these parameters are weakly supervised through the ImV bounding box constraints. This implicit supervision encourages the projected cylinders to tightly enclose pedestrians in each view while maintaining geometric consistency across domains.

As a result, the final training objectives combine from both BEV learning and ImV learning objective functions, as follows:

$$L = L_{cls}^{bev} + L_{KL}^{bev} + L_{smooth\_L1}^{img} + L_{GIoU}^{img}.  \tag{13}$$

To further improve recall in crowded scenes, we introduce an *Adaptive Hungarian Matching* strategy inspired by one-to-many assignment Jia et al. (2023). This allows multiple spatially adjacent and non-overlapping queries to be matched to a single BEV ground-truth point, increasing the number of positive samples and enhancing robustness under heavy occlusion.

## 4 Experiments

### 4.1 Datasets.

Following prior work, we benchmark MVDGC on two well-known multi-view pedestrian detection (MVPD) datasets including WildTrack and MultiViewX. We also further implement additional experiment to evaluate the capability of scene generalization on a large-scale dataset GMVD.

- *WildTrack* Chavdarova et al. (2017) is a real-world outdoor dataset with 7 synchronized and calibrated cameras with a resolution of $1920 \times 1080$. The monitored area spans $12\,\mathrm{m} \times 36\,\mathrm{m}$ and is discretized into a $480 \times 1440$ grid with $2.5\,\mathrm{cm}$ spatial resolution. Each frame contains about 20 pedestrians, each person visible from an average of 3.74 camera views. This dataset comprises 400 frames, where the first 360 frames are used for training, and the last 40 frames are used for testing. Each frame contains 7 images for 7 views.

- *MultiViewX* Hou et al. (2020) is a synthetic counterpart to WildTrack, featuring 6 calibrated cameras. It covers a $16\,\mathrm{m} \times 25\,\mathrm{m}$ area, discretized into a $640 \times 1000$ grid using $2.5\,\mathrm{cm}$ cells. Each frame includes on average around 40 pedestrians, providing a denser populated scenario. This dataset also comprises 400 frames, where the first 360 frames are used for training, and the last 40 frames are used for testing. Each frame contains 6 images for 6 views.

- *GMVD* Vora et al. (2023) is a large-scale synthetic dataset consisting of multiple scenes with diverse camera configurations and ground-plane dimensions. It is specifically designed to evaluate the generalizability of multi-view detection model. Unlike the previously discussed datasets, which are captured within a single, continuous, and relatively constrained environment, GMVD provides variations in both temporal and weather conditions. Moreover, whereas the earlier datasets use a single scene for both training and testing, which may lead to overfitting, the GMVD dataset offers distinct environments for its training and evaluation splits, thereby enabling a more rigorous assessment of cross-scene generalization. The number of scenes for training is 53 (4,983 frames) and the number of scenes for testing is 10 (1,012 frames). Each frame contains from 3 to 8 images due to the varying number of cameras in the setup.

We use the three above datasets to evaluate and compare with other methods for MVPD. We further implement BEV tracking and ImV tracking on WildTrack and MultiViewX, and compare with other methods that reported tracking results.

## 4.2   Evaluation Metrics.

We evaluate MVDGC on two primary tasks: MVPD and BEV-ImV pedestrian detection and tracking.

- For the first task, we using five standard metrics: MODA (Multi-Object Detection Accuracy), MODP (Multi-Object Detection Precision) Kasturi et al. (2008), Precision, Recall, and F1-score. While MODA shows the localization accuracy, MODP accesses the localization precision of detections.

- For the second task, we conduct tracking in BEV and translate the results into 2D views. This task evaluates the temporal and spatial consistency of our method across 2D and 3D domains. We compare against existing single-view tracking methods and adopt five tracking metrics including HOTA, Association Accuracy (AssA), and Detection Accuracy (DetA) from  Luiten et al. (2021), as well as Multi-Object Tracking Accuracy (MOTA), and Identity F1 Score (IDF1) Ristani et al. (2016).

## 4.3   Implementation Details.

Our model is implemented using PyTorch Paszke et al. (2019) and trained/tested on a NVIDIA RTX A6000. We train the model using AdamW optimizer with batch size of 1 and an initial learning rate of $10^{-4}$, decayed through a MultiStepLR schedule after 20 epochs for a total of 30 epochs. The input images are resized to $640 \times 640$. The model adopts a ViT backbone Zong et al. (2023), and follows the Deformable DETR configuration, with 4 encoder layers and 4 decoder layers. We initialize 768 cylindrical queries adaptively and uniformly based on the BEV grid size for every scene. The radius and height is approximated for a human with height of 1.7 m. For query assignment, we adopt adaptive one-to-many Hungarian Matching with $K = 3$, meaning that the three closest predictions are allowed to match the same ground-truth target, provided that their distances remain within a predefined threshold. During inference, keypoint-based NMS with a threshold of 0.6 is applied to remove redundant predictions and obtain the final set of BEV outputs.

## 4.4   Baselines.

We evaluate MVDGC against a comprehensive set of multi-view pedestrian detection and tracking baselines as follows:

• For multi-view pedestrian detection, we include both anchor-based and centralized approaches. The anchor-based category comprises classical methods such as POM-CNN Fleuret et al. (2008), RCNN with clustering Xu & Qiu (2016), DeepMCD Chavdarova & Fleuret (2017), and Deep-Occlusion Baqué et al. (2017). Centralized BEV-based methods include MVDet Hou et al. (2020), SHOT Song et al. (2021), MVDetr Hou & Zheng (2021), 3DROM Qiu et al. (2022), MVAug Engilberge et al. (2023), EarlyBird Teepe et al. (2024a), and MVFP Aung et al. (2024). We additionally compare with methods that leverage extra modalities or temporal cues, such as MVTT Lee et al. (2023), TrackTacular Teepe et al. (2024b), and PVH-Enhanced Alturki et al. (2025).

• For the tracking task, we report results in both the BEV and image views. BEV tracking baselines on MVPD include KSP-DO Chavdarova et al. (2018), KSP-DO-ptrack Chavdarova et al. (2018), GLMB-YOLOv3 Ong et al. (2020), GLMB-DO Ong et al. (2020), DMCT You & Jiang (2020), DMCT-Stack You & Jiang (2020), ReST Cheng et al. (2023), EarlyBird Teepe et al. (2024a), and TrackTacular Teepe et al. (2024b). Since MVDGC is the only method capable of producing both BEV and ImV tracking results, these BEV-only approaches are not directly applicable for ImV tracking. For image-view tracking, we compare against representative CNN-based and Transformer-based trackers, including OCSORT Cao et al. (2023), ByteTrack Zhang et al. (2022), DeepOCSORT Maggiolino et al. (2023), BoT-SORT Aharon et al. (2022), StrongSORT Du et al. (2023), BoostTrack++ Stanojević & Todorović (2024), MOTRv2 Zhang et al. (2023), and MeMOTR Gao & Wang (2023).

Table 2: **Multi-view pedestrian detection (MVPD)** performance on the **WildTrack** and **MultiViewX** benchmarks. Methods marked with [†] and highlighted in `gray`, utilize extra information (Extra Info) such as temporal cues or additional regional proposals, thus, they are only used as reference without direct comparison. The best results are in **bold**, and the second-best are underlined.

| Method | Extra Info | WildTrack | | | | | MultiViewX | | | | |
|---|---|---|---|---|---|---|---|---|---|---|---|
| | | MODP | MODA | Pre | Rec | F1 | MODP | MODA | Pre | Rec | F1 |
| *Anchor-based Methods:* | | | | | | | | | | | |
| POM-CNN  Fleuret et al. (2008) | ✗ | 30.5 | 23.2 | 75 | 55 | 63.5 | – | – | – | – | – |
| RCNN-based  Xu & Qiu (2016) | ✗ | 18.4 | 11.3 | 68 | 43 | 52.7 | 46.4 | 18.7 | 63.5 | 43.9 | 51.9 |
| DeepMCD Chavdarova & Fleuret (2017) | ✗ | 64.2 | 67.8 | 85 | 82 | 83.5 | 73.0 | 70.0 | 85.7 | 83.3 | 84.5 |
| Deep-Occ Baqué et al. (2017) | ✗ | 53.8 | 74.1 | 95 | 80 | 86.8 | 54.7 | 75.2 | 97.8 | 80.2 | 88.1 |
| *Centralized-based Methods:* | | | | | | | | | | | |
| MVDet Hou et al. (2020) | ✗ | 75.7 | 88.2 | 94.7 | 93.6 | 94.1 | 79.6 | 83.9 | 96.8 | 86.7 | 91.5 |
| SHOT Song et al. (2021) | ✗ | 76.5 | 90.2 | 96.1 | 94.0 | 95.0 | 82.0 | 88.3 | 96.6 | 91.5 | 94.0 |
| MVDetr Hou & Zheng (2021) | ✗ | 82.1 | 91.5 | **97.4** | 94.0 | 95.7 | 91.3 | 93.7 | **99.5** | 94.5 | 96.9 |
| 3DROM Qiu et al. (2022) | ✗ | 75.9 | 93.5 | 97.2 | 96.2 | 96.7 | 84.9 | 95.0 | 99.0 | 96.1 | 97.5 |
| MVAug Engilberge et al. (2023) | ✗ | 79.8 | 93.2 | 96.3 | 97.0 | 96.6 | 89.7 | 95.3 | 99.4 | 95.9 | 97.6 |
| Earlybird Teepe et al. (2024a) | ✗ | 81.8 | 91.2 | 94.9 | 96.3 | 95.6 | 90.1 | 94.2 | 98.6 | 95.7 | 97.1 |
| MVFP Aung et al. (2024) | ✗ | 78.8 | 94.1 | 96.4 | 97.7 | 97.0 | 85.1 | 95.7 | 98.4 | 97.2 | 97.8 |
| MVTT[†] Lee et al. (2023) | ✓ | 81.3 | 94.1 | 97.6 | 96.5 | 97.0 | 92.8 | 95.0 | 99.4 | 95.6 | 97.5 |
| TrackTacular[†]Teepe et al. (2024b) | ✓ | 77.5 | 93.2 | 97.3 | 95.8 | 96.5 | 75.0 | 96.5 | 99.4 | 97.1 | 98.2 |
| PVH-Enc[†]Alturki et al. (2025) | ✓ | 82.4 | 93.6 | 96.6 | 97.0 | 96.8 | 95.0 | 97.3 | 99.5 | 97.9 | 98.7 |
| *Query-based Methods:* | | | | | | | | | | | |
| **MVDGC (ours)** | ✗ | **83.8** | **94.2** | **97.4** | 96.9 | **97.1** | **93.0** | **95.7** | 98.8 | 96.9 | **97.8** |

## 4.5 Performance Comparison

### 4.5.1 MVPD Performance

To evaluate the MVPD performance of our method, we adopt the $(x, y)$ coordinates of the predicted $P^{3D}$ as footpoint estimates, in accordance with the ground-truth annotations, which provide only footpoint positions. As shown in Table 2, our method achieves SOTA performance across multiple metrics, MODP, MODA, Precision, and F1 on WildTrack, and MODP, MODA, and F1 on MultiViewX. This improvement stems from the query-based absolute object position prediction and the enforced supervision in both BEV and image views, whereas centralized approaches such as MVDeTr, 3DROM, and Earlybird still suffer from projection distortion. MVFP and PVH attempted to reduce projection distortion by pulling 3D feature. However, due to the estimation via BEV heatmap, they suffer from quantization error and point shifting. Our method performs well on both real-world (WildTrack and MultiviewX) and synthetic datasets (GMVD), with superior localization precision. Notably, our model achieves comparable or better performance when compared with methods that required additional information, such as MVTT (region proposals), TrackTacular (explicit temporal modeling), PVH (object's visual hull).

### 4.5.2 Application of 3D Cylindrical Representation to BEV and ImV Pedestrian Tracking

Tracking objects from a single camera view is inherently difficult due to frequent occlusions. While existing MVPD methods leverage multi-view information to achieve strong detection and tracking performance on the BEV plane, they typically treat BEV localization and ImV detection as two separate tasks. As a result, the tracking IDs established in BEV cannot be propagated back to the image views. In contrast, MVDGC explicitly constructs a BEV-ImV correspondence and enables ImV tracking by transferring the global IDs generated in the BEV domain. This consistent, cross-view identity assignment is crucial for enhancing ImV tracking performance.

In this experiment, we demonstrate the effectiveness of our 3D cylindrical representation in the context of both ImV and BEV pedestrian tracking. Traditional single-view trackers often struggle under occlusion

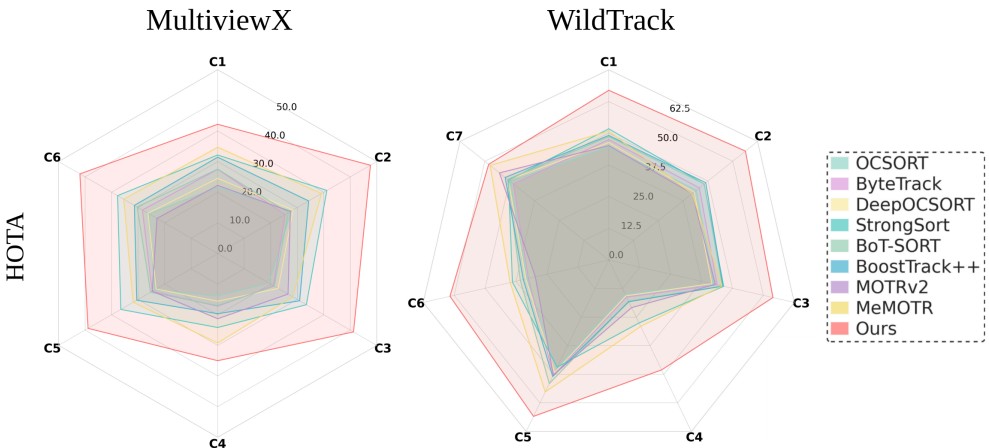

Figure 4: **Per-camera tracking** performance under HOTA on **MultiViewX** (left column figure) and **WildTrack** (right column figure). MultiViewX includes six cameras C1-C6 and WildTrack includes seven cameras C1-C7.

Table 3: **BEV tracking** performance on **WildTrack** dataset. Methods marked with † and highlighted in gray , utilize extra information (Extra Info) such as temporal cues or additional regional proposals, thus, they are only used as reference without direct comparison. The best results are in **bold**, and the second-best are underlined.

| Methods | Extra Info | BEV Tracking on WildTrack | | |
|---|---|---|---|---|
| | | IDF1 ↑ | MOTA ↑ | MOTP ↑ |
| KSP-DO Chavdarova et al. (2018) | ✗ | 73.2 | 69.6 | 61.5 |
| KSP-DO-ptrack Chavdarova et al. (2018) | ✗ | 78.4 | 72.2 | 60.3 |
| GLMB-YOLOv3 Ong et al. (2020) | ✗ | 74.3 | 69.7 | 73.2 |
| GLMB-DO Ong et al. (2020) | ✗ | 72.5 | 70.1 | 63.1 |
| DMCT You & Jiang (2020) | ✗ | 77.8 | 72.8 | 79.1 |
| DMCT Stack You & Jiang (2020) | ✗ | 81.9 | 74.6 | 78.9 |
| ReST Cheng et al. (2023) | ✗ | 85.7 | 81.6 | – |
| EarlyBird Teepe et al. (2024a) | ✗ | 92.3 | 89.5 | **86.6** |
| TrackTacular† Teepe et al. (2024b) | ✓ | 95.3 | 91.8 | 85.4 |
| **MVDGC** | ✗ | **95.8** | **92.1** | 84.9 |

due to the lack of geometric reasoning, while multi-view tracking methods, despite leveraging different perspectives, typically produce occupancy maps in BEV space without a mechanism to associate them with 2D image-space detections, making them incompatible with this downstream task. It is worth noting that the primary objective of this experiment is not to develop a new tracking method, but to evaluate the properties of the proposed representation in a tracking setting. Our tracking experiments are therefore intended as secondary validation of the proposed 3D cylindrical representation and its dual BEV-ImV geometric constraints. Specifically, they validate (i) the ability of our representation to preserve geometric correspondence and consistent identities across BEV and image domains, and (ii) the quality of the underlying detections through a tracking-by-detection framework. Our method effectively bridges this gap by a 3D cylindrical representation that is jointly supervised in both BEV and image space. Each 3D cylinder, which encodes pedestrian's BEV point along with its bounding box in image space, enabling seamless projection between two domains. This unified representation naturally mitigates occlusion by maintaining a consistent global identity across camera views. To enable single-view tracking, we integrate our BEV detection model with a standard tracking-by-detection pipeline using association technique such as ByteTrack Zhang et al. (2022). Associations are performed based on the Euclidean distance between pedestrian coordinates in BEV

Table 4: **ImV tracking** performance on **WildTrack** dataset. The best results are in **bold**, and the second-best are underlined.

| Methods | Global | ImV Tracking on WildTrack | | | | |
|---------|--------|------|------|------|------|------|
| | ID | HOTA | MOTA | IDF1 | AssA | DetA |
| *CNN based:* | | | | | | |
| OCSORT Cao et al. (2023) | ✗ | 41.9 | 33.3 | 52.6 | 48.3 | 36.6 |
| ByteTrack Zhang et al. (2022) | ✗ | 42.3 | 24.7 | 52.0 | 45.9 | 39.8 |
| DeepOCSORT Maggiolino et al. (2023) | ✗ | 42.2 | 29.3 | 52.2 | 49.0 | 36.5 |
| BoT-SORT Aharon et al. (2022) | ✗ | 45.0 | 32.1 | 55.4 | 49.8 | 41.2 |
| StrongSORT Du et al. (2023) | ✗ | 45.7 | 37.1 | 59.3 | 53.1 | 39.6 |
| BoostTrack++ Stanojević & Todorović (2024) | ✗ | 45.1 | 34.6 | 57.4 | 54.4 | 37.7 |
| *Transformer based:* | | | | | | |
| MOTRv2 Zhang et al. (2023) | ✗ | 43.4 | 23.9 | 50.1 | 57.3 | 33.9 |
| MeMOTR Gao & Wang (2023) | ✗ | 48.8 | 30.8 | 57.0 | 62.1 | 39.2 |
| **MVDGC** | ✓ | **65.4** | **72.9** | **86.0** | **72.9** | **60.6** |

Table 5: **BEV tracking** performance on **MultiViewX** dataset. Methods marked with † and highlighted in gray , utilize extra information (Extra Info) such as temporal cues or additional regional proposals, thus, they are only used as reference without direct comparison. The best results are in **bold**, and the second-best are underlined.

| Methods | Extra | BeV Tracking on MultiViewX | | |
|---------|-------|------|------|------|
| | Info | IDF1↑ | MOTA↑ | MOTP↑ |
| EarlyBird Teepe et al. (2024a) | ✗ | 82.4 | 88.4 | **86.2** |
| TrackTacular† Teepe et al. (2024b) | ✓ | 83.4 | 91.8 | 84.7 |
| **MVDGC** | ✗ | **84.1** | **92.1** | 85.3 |

space $(x, y)$. Once BEV-space tracklets are formed, each cylinder is assigned by a unique identity, which is then propagated to corresponding 2D bounding boxes across all camera views through camera projection.

As reported in Tables 3 and 5, when transferring the pedestrian detection results in BEV space from Table 2, our method achieves state-of-the-art performance on both datasets. Specifically, MVDGC attains the highest IDF1 scores (95.8 and 84.1) and MOTA scores (92.1 and 92.1) on WildTrack and MultiViewX, respectively. Notably, despite the adoption of an off-the-shelf tracker, our approach outperforms EarlyBird and TrackTacular, which respectively incorporate a dedicated re-identification branch and exploit temporal cues from previous frames. For ImV tracking, MVDGC is the only multi-view pedestrian detection framework capable of producing 2D bounding boxes directly from a unified 3D cylindrical representation. Leveraging the globally consistent identities established during BEV tracking, our method achieves superior ImV tracking performance. As shown in Tables 4 and 6, MVDGC surpasses both CNN-based and Transformer-based ImV tracking methods across all evaluation metrics by a large margin. For instance, compared to MeMOTR Gao & Wang (2023), our MOTA improves by 42.1 points on WildTrack and by 9.6 points on MultiViewX. Although MultiViewX poses greater challenges due to its denser pedestrian distribution, MVDGC consistently maintains superior tracking performance, demonstrating strong robustness in crowded scenes.

Figure 4 presents a per-camera comparison of tracking performance in terms of HOTA on the MultiViewX and WildTrack datasets. Each radar plot reports results across individual camera views, providing a fine-grained analysis of cross-view tracking robustness. Our method consistently achieves the highest HOTA scores across all cameras on both datasets, demonstrating stable and balanced tracking performance regardless of viewpoint. In contrast, competing methods exhibit larger performance fluctuations across cameras, indicating increased sensitivity to occlusion, and viewpoint changes. These results highlight the robustness of tracklet

Table 6: **ImV tracking** performance on **MultiViewX** dataset. The best results are in **bold**, and the second-best are underlined.

| Methods | Global | ImV Tracking on MultiViewX | | | | |
|---|---|---|---|---|---|---|
| | ID | HOTA | MOTA | IDF1 | AssA | DetA |
| *CNN based:* | | | | | | |
| OCSORT Cao et al. (2023) | ✗ | 22.7 | 17.3 | 25.0 | 21.6 | 23.9 |
| ByteTrack Zhang et al. (2022) | ✗ | 24.3 | 25.0 | 25.8 | 12.7 | 48.0 |
| DeepOCSORT Maggiolino et al. (2023) | ✗ | 23.8 | 18.1 | 26.7 | 22.52 | 25.4 |
| BoT-SORT Aharon et al. (2022) | ✗ | 25.1 | 23.2 | 26.3 | 13.5 | 47.6 |
| StrongSORT Du et al. (2023) | ✗ | 34.9 | 29.7 | 41.1 | 37.2 | 32.8 |
| BoostTrack++ Stanojević & Todorović (2024) | ✗ | 30.0 | 27.9 | 38.5 | 31.5 | 28.8 |
| *Transformer based:* | | | | | | |
| MOTRv2 Zhang et al. (2023) | ✗ | 24.7 | 12.3 | 28.7 | 25.5 | 24.2 |
| MeMOTR Gao & Wang (2023) | ✗ | 33.4 | 30.4 | 37.1 | 35.5 | 31.6 |
| **MVDGC** | ✓ | **48.4** | **40.0** | **61.8** | **50.5** | **48.1** |

transfer enabled by our approach, and verify the effectiveness of the proposed 3D cylindrical query as a strong BEV-ImV bridge for maintaining global identities across views.

## 4.6 Qualitative Results

Figure 5 illustrates the alignment between the predicted 3D cylinders and their corresponding 2D bounding box in image views in real-world WildTrack dataset. The cylinder projections consistently and tightly enclose each pedestrian in camera views, resulting in the variation in radius and height of cylinders in 3D space. This demonstrates our model's awareness about 3D occupancy and spatial context, along with the consistency in 3D and 2D predictions. Figure 6 depicts the predicted 3D cylinders in synthetic MultiViewX dataset with more object appearance. Even in crowded scenes with significant occlusion, the method maintains accurate spatial alignment and robust localization across views, highlighting its effectiveness in challenging multi-view scenarios.

Figure 7 presents a qualitative comparison of ground-plane predictions produced by our method and recent baselines MVDeTr, 3DROM, and EarlyBird on the WildTrack and MultiViewX datasets. These baselines are selected based on the availability of reproducible source code and pretrained checkpoints. Among open-source methods, 3DROM and MVDeTr report the highest MODA and MODP scores, respectively. Our predictions (•) align more closely with the ground truth (•), consistent with the superior MODP reported in Table 2. In contrast, 3DROM exhibits numerous low-precision detections (red) (low MODP score), while MVDeTr suffers from a substantial number of false negatives (black) (low MODA score). To better understand the strengths of MVDGC, we investigate two primary failure cases observed in prior methods: dense crowding in central regions and localization near scene boundaries.

- In crowded central areas, pedestrians frequently occlude each other across multiple views, causing centralized-based methods (MVDeTr, 3DROM, EarlyBird) to produce imprecise and misaligned predictions. For example, on the WildTrack dataset, MVDeTr struggles to precisely locate objects in the middle of the scene. A fundamental reason is that these methods rely heavily on foot-point features projected onto the BEV plane. In crowded scenes, the feet of nearby pedestrians overlap and occlude each other, producing ambiguous and entangled features at the ground level. This problem is further exacerbated by perspective projection, which compresses overlapping pedestrians into similar BEV locations on the dense heatmap, making it even more difficult to distinguish individual ground points. MVDGC avoids this limitation by sampling features from the full body extent of each pedestrian through the cylindrical projection. When foot regions are occluded, features from

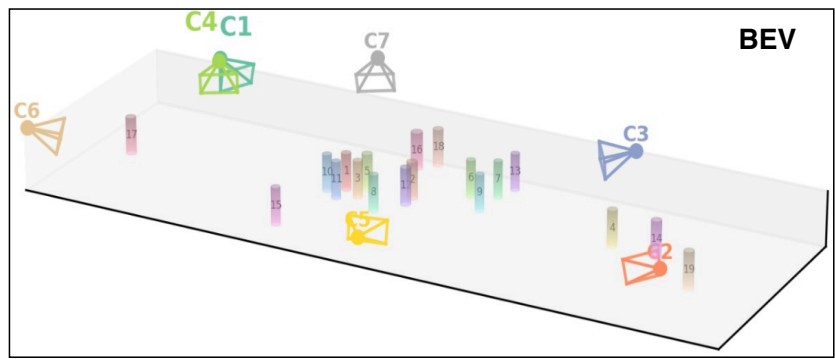

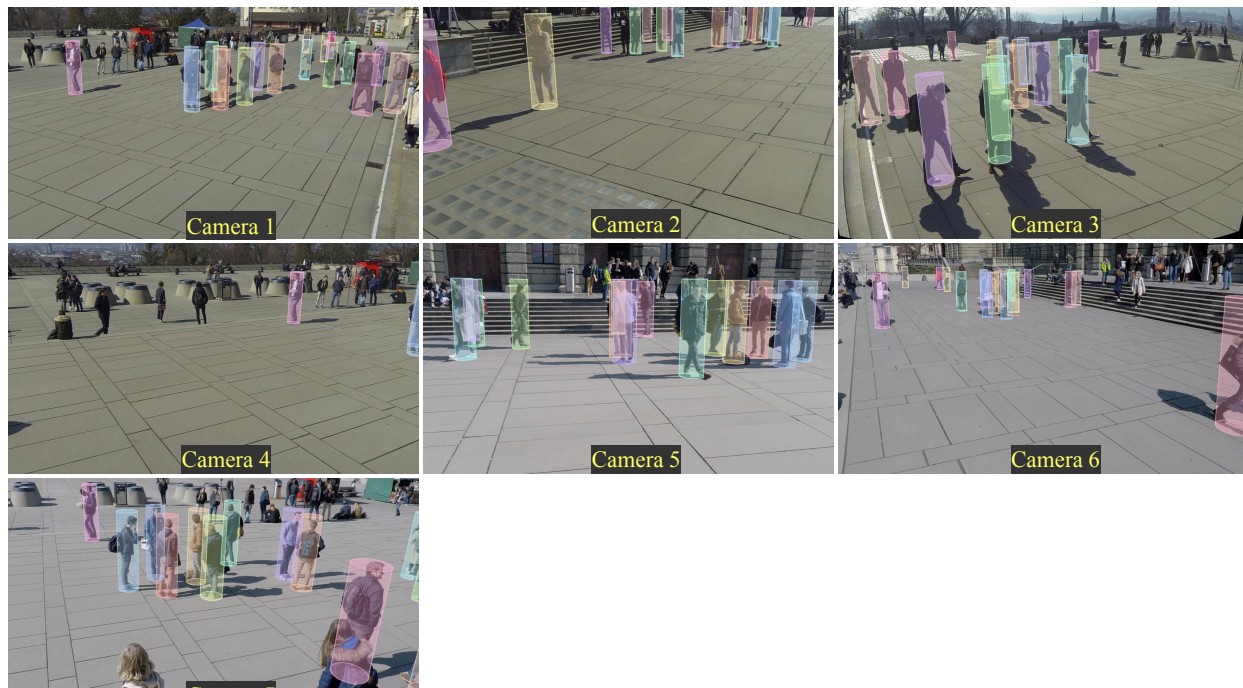

Figure 5: Visualization of predicted cylinders in 3D space (top) and the corresponding projections on 2D camera views (bottom) for **WildTrack** dataset. Each object is identified by a unique color and ID.

the torso, shoulders, or head still contribute to the localization, providing complementary cues that help resolve the pedestrian's ground-plane position even under heavy lower-body occlusion.

- At scene borders, objects may only be clearly visible from one or two cameras, or appear too small in distant views to be reliably detected. 3DROM exhibits several low-precision predictions near the border. Although its random occlusion augmentation strategy improves robustness in central regions, 3DROM does not address the low precision at the periphery. EarlyBird shows both false negatives and false positives near scene borders, where insufficient multi-view overlap leads to ambiguous feature aggregation on the BEV plane.

MVDGC mitigates both failure modes through its object-centric design. For crowded regions, the cylindrical queries avoid the BEV feature aggregation problem. Each query independently regresses its position rather than competing for activation on a shared dense map. For border regions, the Multi-view Adaptive Fusion module (Section 3.5) learns instance-specific attention weights that can prioritize the one or two views where the pedestrian is clearly visible, rather than uniformly averaging features from all cameras including those with poor visibility.

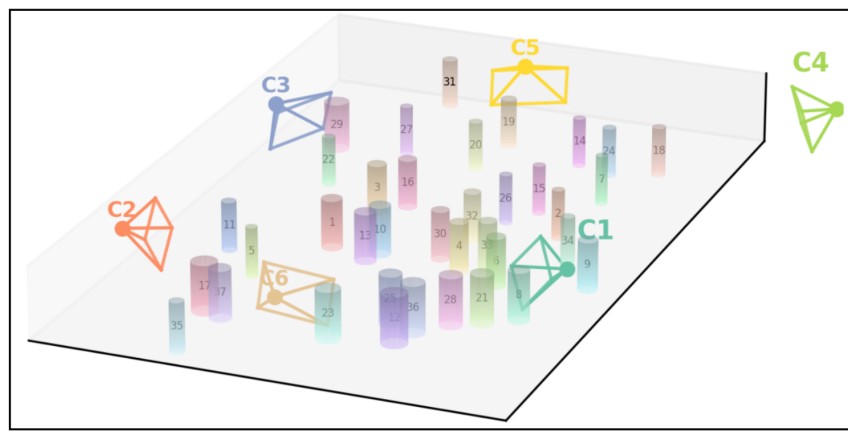

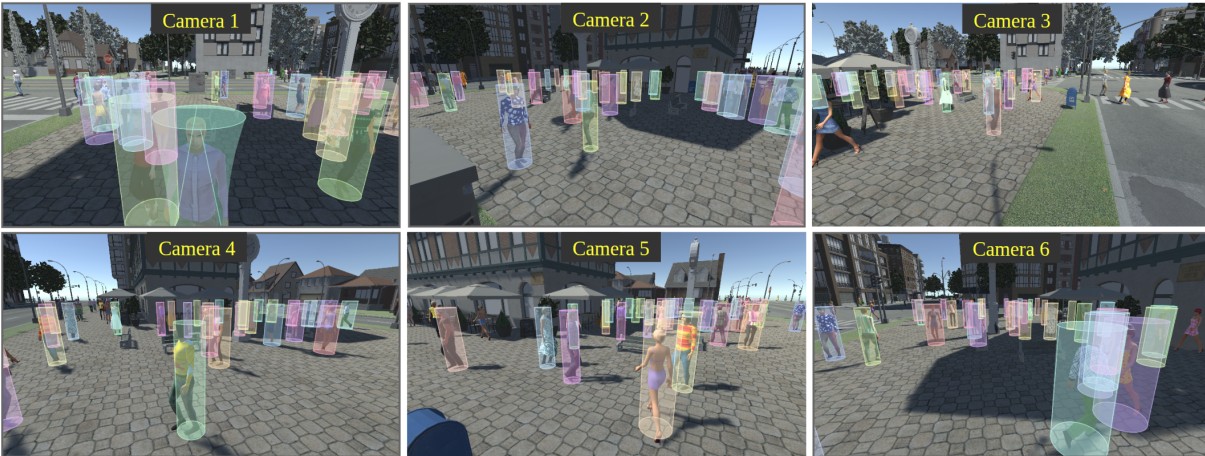

Figure 6: Visualization of predicted cylinders in 3D space (top) and the corresponding projections on 2D camera views (bottom) for **MultiViewX** dataset. Each object is identified by a unique color and ID.

### 4.7 Ablations Study

**Scene Generalization** Although WildTrack and MultiViewX are widely used benchmarks for MVPD, both datasets are limited in scale. Each of them contains only 400 continuous frames captured within a single scene. As a result, the camera setups and environmental conditions in their training and testing splits largely overlap, restricting their ability to evaluate generalization. In contrast, the GMVD dataset spans multiple scenes with diverse environmental conditions and camera configurations, and its training and testing partitions are fully disjoint. This makes GMVD a more suitable benchmark for assessing the scene-level generalization capability of MVPD approaches. Table 7 presents the detection results of baselines including MVDet, GMVD, MVFP, and our proposed MVDGC on GMVD. MVDGC is designed to accommodate an arbitrary number of camera views, and its Multi-view Feature Sampling module aggregates object-centric appearance features guided by geometric projections, which helps reduce sensitivity to background clutter and environmental variation. As a result, MVDGC achieves the highest MODP and maintains consistently strong performance across all remaining metrics, demonstrating robustness to geometric variability, heterogeneous camera layouts, and complex scene dynamics.

**Contribution of individual modules.** Table 8 shows that while inter- or intra-interaction alone yields modest gains, their combination significantly improves performance to 90.9 MODA. Adding the Multi-view Adaptive Fusion module alone also boosts metrics, reaching 90.3 MODA. Integrating all three components results in the largest improvement, highlighting their complementary benefits.

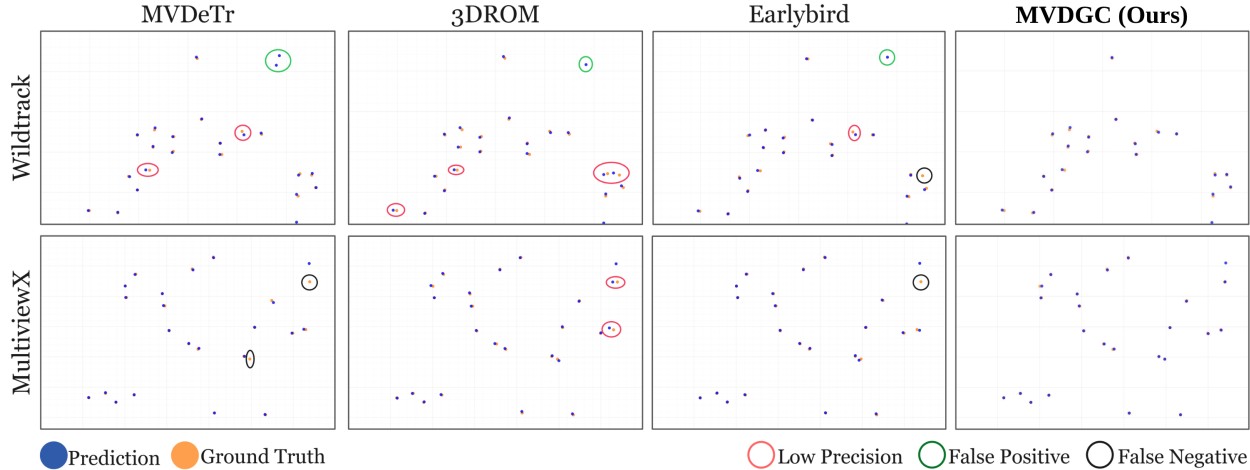

Figure 7: Comparison of ground-plane predictions on the BEV domain across previous methods and MVDGC on **WildTrack** and **MultiViewX** datasets.

Table 7: **Ablation study on scene generalization** on the **GMVD** benchmark. The best results are in **bold**, and the second-best are underlined.

| Method | Scene Generalization on GMVD | | | | |
|---|---|---|---|---|---|
| | MODP | MODA | Precision | Recall | F1 |
| *Centralized-based Methods:* | | | | | |
| MVDet Hou et al. (2020) | 72.8 | 50.5 | 83.6 | 64.7 | 72.9 |
| GMVD Vora et al. (2023) | 76.3 | 68.2 | 91.5 | 75.5 | 82.7 |
| MVFP Aung et al. (2024) | 76.5 | **73.3** | **93.0** | **79.2** | **85.5** |
| *Query-based Methods:* | | | | | |
| **MVDGC** | **79.6** | 71.2 | **93.0** | 76.9 | 84.1 |

**Contribution of losses.** The equations of BEV loss and ImV loss are mentioned in Equation 13. Table 9 demonstrates that ImV supervision leads to substantial improvements over BEV-only supervision, particularly in MODP. This highlights the advantage of accumulating supervision from multiple image views. The best performance is achieved with joint BEV-ImV supervision, emphasizing the importance of dual-space consistency and resolving the limitations of centralized-based methods. In terms of regression loss, Table 10 shows both L1 and Smooth L1 losses perform competitively, with Smooth L1 slightly outperforming L1 in all metrics. KL loss performs best for point regression, thanks to its spatial uncertainty modeling, which handles annotation ambiguity, enhances precision, and prevents overfitting. These findings highlight the advantage of modeling uncertainty explicitly in spatial representations, especially under ambiguous supervision scenarios.

**Assignment strategies.** Table 11 compares several assignment strategies and highlights the limitations of standard one-to-one Hungarian matching, which results in low recall and a reduced MODA score (76.5). Allowing multiple queries to associate with the same ground truth, as done in KNN one-to-many matching, alleviates this issue by improving recall and yielding more stable optimization. Motivated by these observations, we introduce Adaptive Hungarian Matching, which dynamically controls how many queries may be assigned to each ground-truth object based on geometric proximity. This prevents redundant matches from spatially adjacent objects and leads to the best overall performance. Table 12 further analyzes the role of the parameter $K$ in controlling the maximum number of matched queries. The results show that $K = 3$ provides the most effective balance between flexibility and precision, achieving the strongest performance across all evaluation metrics. Together, these two tables demonstrate that both the matching strategy and the choice of $K$ are crucial for achieving robust and accurate multi-view pedestrian detection.

Table 8: Ablation study on the effect of the proposed modules on the **WildTrack** dataset.

| Intra Interaction | Inter Interaction | Adaptive Fusion | MODA | MODP | F1 |
|:---:|:---:|:---:|:---:|:---:|:---:|
| ✗ | ✗ | ✗ | 88.7 | 81.2 | 94.5 |
| ✓ | ✗ | ✗ | 89.4 | 79.6 | 94.7 |
| ✗ | ✓ | ✗ | 88.3 | 82.1 | 94.3 |
| ✗ | ✗ | ✓ | 90.3 | 81.0 | 95.1 |
| ✓ | ✓ | ✗ | 90.9 | 81.3 | 95.4 |
| ✓ | ✓ | ✓ | **94.2** | **83.1** | **97.1** |

Table 9: Ablation study on dual geometric constraints on the **WildTrack** dataset.

| $\mathcal{L}_{\text{BEV}}$ | $\mathcal{L}_{\text{img}}$ | MODA | MODP | $F_1$ |
|:---:|:---:|:---:|:---:|:---:|
| ✓ | ✗ | 85.7 | 72.7 | 92.6 |
| ✗ | ✓ | 89.6 | 82.1 | 94.9 |
| ✓ | ✓ | **94.2** | **83.1** | **97.1** |

Table 10: Ablation study on different types of losses on the **WildTrack** dataset.

| **Loss** | MODA | MODP | $F_1$ |
|:---|:---:|:---:|:---:|
| L1 | 93.8 | 81.6 | 96.9 |
| Smooth L1 | 93.9 | 82.6 | 97.0 |
| **KL Loss (ours)** | **94.2** | **83.1** | **97.1** |

Table 11: Ablation study on various assignment strategies on the **WildTrack** dataset.

| **Strategy** | MODA | MODP | $F_1$ |
|:---|:---:|:---:|:---:|
| Hungarian | 76.5 | 82.6 | 88.9 |
| K-nearest neighbour | 91.8 | 82.2 | 95.9 |
| **Adaptive Hungarian (ours)** | **94.2** | **83.1** | **97.1** |

Table 12: Ablation study on the number of assignments on the **WildTrack** dataset.

| **K** | MODA | MODP | $F_1$ |
|:---|:---:|:---:|:---:|
| 1 | 89.3 | 81.5 | 93.9 |
| 2 | 92.6 | 82.2 | 96.4 |
| **3 (ours)** | **94.2** | **83.1** | **97.1** |
| 4 | 93.7 | 83.3 | 96.8 |

Table 13: Ablation study on various query types on the **WildTrack** dataset.

| **Query type** | MODA | MODP | $F_1$ |
|:---|:---:|:---:|:---:|
| 3D reference point | 82.9 | 70.7 | 91.2 |
| **3D cylindrical query (ours)** | **94.2** | **83.1** | **97.1** |

Table 14: Ablation study on the number of queries on the **WildTrack** dataset.

| **# queries** | MODA | MODP | $F_1$ |
|:---|:---:|:---:|:---:|
| 256 | 91.3 | 80.4 | 93.2 |
| 512 | 92.9 | 81.9 | 95.8 |
| **768 (ours)** | **94.2** | **83.1** | **97.1** |
| 1024 | 93.5 | 83.5 | 96.2 |

**Query configurations.** The query configuration used in our framework is illustrated in Figure 3 and detailed in Section 4.3. Table 13 compares the use of 3D reference points against our proposed 3D cylindrical queries. Notably, conventional 3D bounding box queries are not directly applicable in the multi-view pedestrian detection setting for three reasons: (1) existing datasets do not provide full 3D box annotations; (2) projecting a 3D box into image views produces non-uniform polygons rather than rectangles, requiring additional post-processing; and (3) this added complexity does not translate to measurable performance gains. In contrast, simple 3D reference-point queries fail to leverage the geometric constraints encoded in 2D bounding boxes, resulting in only 82.9 MODA. The strong performance of the cylindrical queries demonstrates the effectiveness of our design, which is specifically tailored to operate under weak 3D supervision and the annotation limitations common in MVPD datasets. Furthermore, Table 14 shows that distributing 768 cylindrical queries uniformly across the BEV plane yields the best results, confirming the importance of both the query structure and density in achieving robust detection performance.

## 5 Conclusion

In this work, we present MVDGC, a 3D cylindrical query-based framework for multi-view pedestrian detection that unifies ground-plane/Bird's-Eye View (BEV) localization and image-view detection within a single geometric representation. Unlike anchor-based methods, which rely on late-stage association of independent 2D detections and therefore struggle with occlusion and weak cross-view reasoning, MVDGC performs early feature alignment through geometry-aware queries, enabling coherent spatial reasoning across views. In contrast to centralized-based approaches that depend on dense BEV feature maps and suffer from perspective distortion and quantization errors, our method directly regresses absolute object attributes by sampling features from raw image views, effectively eliminating BEV distortion. The proposed cylindrical representation establishes an explicit and consistent bridge between BEV ground points and ImV bounding boxes, providing a shared global identity across spatial domains and time. Extensive experiments and ablation studies demonstrate that this BEV-ImV consistency leads to superior localization accuracy and robustness, and further benefits downstream tasks such as image-based tracking, where MVDGC significantly outperforms existing single-view tracking baselines under heavy occlusion.

**Dicussion** A current limitation of our model is its relatively lower recall, as shown in Table 2. This issue primarily arises in scenarios where pedestrians stand closely together, creating severe occlusions that are difficult to resolve even with multiple camera viewpoints. Prior work has addressed this challenge by incorporating additional depth cues: MVFP Aung et al. (2024) and PVH Alturki et al. (2025) employ 3D feature pulling to better distinguish heavily occluded pedestrians. TrackTacular Teepe et al. (2024b) approaches the problem from a temporal perspective by leveraging past information to recover missing detections. For future work, we plan to investigate the use of depth-generation models to synthesize point-cloud-like representations, which may help improve detection robustness in highly occluded scenes.

**Acknowledgments** This material is based upon work supported by the National Science Foundation (NSF) under Award 2443877 and Aviagen Inc.

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
