# OpenReview forum: "MVDGC: Joint 3D and 2D Multi-view Pedestrian Detection via Dual Geometric Constraints"
_TMLR — Accepted by TMLR_

### Review · Reviewer_PaA7 · 2026-03-10

**Summary Of Contributions:**

Contributions:
1. The paper proposes MVDGC, a framework that jointly estimates pedestrian locations in BEV and 2D bounding boxes in ImV.
2. The paper introduces the 3D cylindrical representation for the MVPD task, which addresses the bottleneck in the existing anchor-based and centralized-based methods.
3. The paper provides comprehensive experimental results and ablation studies on three datasets to demonstrate the effectiveness of pedestrian localization and tracking with the proposed method.

Strength:
1. The paper has a clear structure and is easy to follow.
2. The paper proposes a different approach to solve the limitations of the existing approaches on the MVPD task.
3. The experiments are comprehensive with adequate ablations.

Weakness:
1. The improvement over the MVPD (Tab. 2) and BEV tracking (Tab. 3 and Tab. 5) is marginal.
2. Some details of the method is not clear. For example, in the decoding part, how to initialize the sparse set of 3D queries (how to set the initial $P^{3D}$? How does the model output the predicted variance $\sigma$ in equ. (9) and the foreground probability $p_f$ in equ. (10)? See more details in the Requested Changes section.

**Audience:**

Yes

**Audience Explanation:**

The paper provides a novel angle to address the MVPD task with the 3D cylindrical representation. It potentially benefits the researchers of pedestrian detection and tracking, and also the researchers working on related BEV-ImV alignment.

**Broader Impact Concerns:**

N.A.

**Claims And Evidence:**

Yes

**Claims Explanation:**

The paper first compares and distinguishes the proposed 3D cylindrical representation with prior work, and then introduces the details of the proposed methods. Finally, extensive experiments and ablations on three datasets show the effectiveness of the proposed approach. The claims are well-supported by accurate, convincing and clear evidence.

**Requested Changes:**

Please clarify and address the Weakness 2 mentioned above. Also, there are some minor problems that require the clarification/revision from the authors:
1. In Sec. 3.3,  the authors inject 3D positional encodings into $F^{3D}$. Why do we need the 3D positional encodings for the features? The position information is already encoded in $P^{3D}$. If you add the positional encoding, then the later operations will not be permutation equivariant anymore, i.e., change the order of the input queries would lead to different detection outputs other than simple permutation.
2. For the predicted spatial offsets $\Delta P^{3D}$, what if the initial pose is too far away from the target person, and the corresponding 2D features is not meaningful for the pose update?
3. For the cylinder decoding (Sec. 3.6), how do we know if a cylinder does not corresponds to an object, i.e., how to deal with different number of detections for different scenes?
4. Currently Fig. 7 is not clear enough, as the projected points on 2D may overlap. It is hard for the readers to distinguish predictions and ground truths. A better way is to provide the qualitive comparison with the baselines in ImV as Fig. 5 and Fig. 6, and highlight the difference.
5. In the Tab.1 the right-most column, the horizontal lines for each cell are missing.

---

> ### Author Response · Authors · 2026-04-08
> **Rebuttal by Authors (1)**
>
> We sincerely thank the reviewer for the detailed and constructive review. We are grateful for the recognition of our paper's clear structure, the novelty of the 3D cylindrical representation in addressing limitations of existing anchor-based and centralized-based methods, and the comprehensiveness of our experiments and ablation studies. We are also encouraged that the reviewer finds our claims well-supported by accurate and convincing evidence. We address each concern below.
>
> ## **Weakness #1:**
> The improvement over the MVPD (Tab. 2) and BEV tracking (Tab. 3 and Tab. 5) is marginal.
>
> ## **Response:**
> We thank the reviewer for this observation. While some improvements may appear modest when viewed on individual metrics, we would like to emphasize that our method aims for consistency across metrics (especially key metrics in **Multi-view** pedestrian detection task such as MODA, MODP, precision and recall), which is a stronger indicator of robustness than isolated improvements on a single metric. To illustrate this, we compare against the two top-performed methods from Table 2:
>
> **Results on WildTrack:**
>
> | Method | MODP | MODA | Pre | Rec | Avg |
> |-|-|-|-|-|-|
> | MVDeTr | 82.1 (-1.7) | 91.5 (-2.7) | **97.4** (0.0) | 94.0 (-2.9) | 91.3 (-1.8) |
> | MVFP | 78.8 (-5.0) | 94.1 (-0.1) | 96.4 (-1.0) | **97.7** (+0.8) | 91.8 (-1.3) |
> | **MVDGC** | **83.8** | **94.2** | **97.4** | 96.9 | **93.1** |
>
> **Results on MultiViewX:**
>
> | Method | MODP | MODA | Pre | Rec | Avg |
> |-|-|-|-|-|-|
> | MVDeTr | 91.3 (-1.7) | 93.7 (-2.0) | **99.5** (+0.7) | 94.5 (-2.4) | 94.8 (-1.3) |
> | MVFP | 85.1 (-7.9) | **95.7** (0.0) | 98.4 (-0.4) | **97.2** (+0.3) | 94.1 (-2.0) |
> | **MVDGC** | **93.0** | **95.7** | 98.8 | 96.9 | **96.1** |
>
> For clarity, we compute a simple average of 4 above metrics, denoted Avg, to reflect the overall balance in multi-view performance. As shown in the two tables, prior MVPD methods exhibit trade-offs across metrics. For instance, MVFP achieves strong MODA and recall but at the cost of a noticeable drop in MODP on both benchmarks (e.g., 78.8 versus 83.8 on WildTrack), indicating less precise localization. Conversely, MVDeTr attains the highest precision but suffers from lower recall, suggesting missed detections. In contrast, our method achieves consistently strong performance across all metrics. On WildTrack, MVDGC improves both MODP (83.8) and MODA (94.2) while maintaining high precision (97.4) and recall (96.9), resulting in the best overall average (93.1). A similar trend is observed on MultiViewX, where MVDGC achieves the highest Avg score (96.1) with balanced gains across localization and detection metrics. We believe this consistent improvement across complementary metrics reflects more reliable localization, better detection completeness, and overall robustness of the proposed framework.
>
> Regarding BEV tracking, we would like to highlight that MVDGC achieves the highest IDF1 and MOTA on both WildTrack and MultiViewX (Table 3 and Table 5) using only an off-the-shelf tracker (ByteTrack with Euclidean distance association) without any appearance features, re-identification modules, or temporal modeling. For instance, EarlyBird incorporates a dedicated re-identification branch leveraging object appearance features, and TrackTacular explicitly exploits temporal information from previous frames. Our method achieves better results with a significantly simpler tracking pipeline, demonstrating that the detection quality and geometric consistency of our cylindrical representation are sufficient to drive strong tracking performance without task-specific optimization.

---

> ### Author Response · Authors · 2026-04-08
> **Rebuttal by Authors (2)**
>
> ## **Weakness #2:**
> Some details of the method is not clear. For example, in the decoding part, how to initialize the sparse set of 3D queries (how to initialize the sparse set of 3D queries (how to set the initial $P^{3D}$)? How does the model output the predicted variance $\sigma$ in equ. (9) and the foreground probability $p_f$ in equ. (10)?
>
> ## **Response:**
> We thank the reviewer for this question and apologize for the missing details. We address it in two parts:
>
> **Weakness #2.1:**
> In the decoding part, how to initialize the sparse set of 3D queries?
>
> **Response:**
> We have provided the general initialization in Section 3.2 (Cylindrical Query Formulation and Initialization) and the specific parameters in Section 4.3 (Implementation Details). To address the reviewer's concern, we extend Section 4.3 here with a more detailed explanation and will include this in the revised paper.
>
> *"The initial position component $P^{3D}$ = {x, y, h, r} is initialized as follows. The ground-plane coordinates $(x, y)$ are scattered in uniform distribution across the surveillance area based on the true physical dimensions of the scene. Specifically, $N_q=768$ ground points are sampled on a 2D uniform grid spanning the width and length of the BEV ground plane. The shape parameters are initialized using human-size heuristics. The height and radius are randomly sampled with mean values of $h=1.7m$ and $r=0.3m$ with a standard deviation of $\pm 0.1$ for both, approximating the average physical dimensions of a pedestrian.''*
>
> **Weakness #2.2:**
> How does the model output the predicted variance $\sigma$ in equ. (9) and the foreground probability $p_f$ in equ. (10)?
>
> **Response:**
> We clarify how each predicted quantity is obtained and will add the following to the revised paper in Section 3.5.
>
> *Predicted variance $\sigma$ (Equation 9):* The variance $\sigma$ is predicted by a dedicated regression head applied to the refined 3D features $\hat{F}^{3D}$ (from Equation 6). Specifically, the regression head is a lightweight MLP that outputs the predicted variance $\sigma$:
>
> $$\sigma = MLP(\hat{F}^{3D})$$
>
> *Predicted foreground probability $p_f$ (Equation 10):* The foreground probability $p_f$ is produced by a separate classification head applied to the same refined 3D features $\hat{F}^{3D}$, consisting of a linear layer followed by a sigmoid activation:
> $$p_f = Sigmoid(\text{Linear}(\hat{F}^{3D}))$$
>
> ## **Requested Change #1:**
> In Sec. 3.3, the authors inject 3D positional encodings into $F^{3D}$. Why do we need the 3D positional encodings for the features? The position information is already encoded in $P^{3D}$. If you add the positional encoding, then the later operations will not be permutation equivariant anymore, i.e., change the order of the input queries would lead to different detection outputs other than simple permutation.
>
> ## **Response:**
> We thank the reviewer for the question. As mentioned in Section 4.3, our framework is query-based and inspired by Deformable DETR. The use of positional encodings here follows standard practice in DETR-style detectors (e.g., DAB-DETR, Deformable DETR), where positional information is explicitly injected into the query features to guide attention.
>
> In our formulation, each query is defined as $Q^{3D} = (F^{3D}, P^{3D})$ , where $F^{3D}$ is the feature component and $P^{3D}$ is the position component representing a 3D cylindrical query. As stated in Section 3.2, *"during training, $F^{3D}$ is updated through multi-view feature aggregation managed by the three subsequent modules in each decoder layer, and then used to predict the alignment and refine $P^{3D}$"*. In other words, $P^{3D}$ is kept intact throughout the feature aggregation process within each decoder layer, and is mainly used to compute projected 2D reference points that guide multi-view feature sampling through camera projection (Equation 1). The position component $P^{3D}$ is only updated at the end of each decoder layer through the predicted spatial offsets $\Delta P^{3D}$ (Equation 7-8). Without injection, it is not directly accessible to the self-attention module. Therefore, injecting positional encodings derived from $P^{3D}$ into $F^{3D}$ is necessary to make spatial information available to the self-attention module, where queries need to reason about their relative positions to avoid collapsing toward similar visual representations.
>
> Regarding permutation properties, each decoder query in our framework independently predicts whether it corresponds to a real pedestrian and where it is located, based solely on each query's own semantic and geometric identity. Since the final output is an unordered set of detections after confidence filtering, and Hungarian matching during training assigns each query to a target based on content rather than order, the detection results do not depend on the ordering of input queries. In other words, reordering the input queries does not affect the final detection outcomes.

---

> ### Author Response · Authors · 2026-04-08
> **Rebuttal by Authors (3)**
>
> ## **Requested Change #2:**
> For the predicted spatial offsets $\Delta P^{3D}$, what if the initial pose is too far away from the target person, and the corresponding 2D features is not meaningful for the pose update?
>
> ## **Response:**
> We thank the reviewer for this question. This concern is addressed through three complementary mechanisms.
> - First, we use an redundant set of queries uniformly distributed in BEV, ensuring dense coverage so that at least a subset of queries are initialized close to each target. For each object, there exist queries that are already reasonably close to the target, reducing reliance on large corrections from poorly initialized queries.
> - Second, even for misaligned queries, the deformable cross-attention mechanism samples features around the projected reference points with learnable offsets, enabling the model to attend beyond the exact projection location and capture informative contextual cues from a broader spatial region. In addition, self-attention enables interaction between queries, allowing poorly initialized queries to benefit from neighboring queries that are better aligned.
> - Third, the decoder performs iterative coarse-to-fine refinement, where early layers make coarse corrections and later layers progressively refine the pose as alignment improves.
>
> Together, these mechanisms make the pose update robust even when the initial query is not well aligned with the target.
>
> ## **Requested Change #3:**
> For the cylinder decoding (Sec. 3.6), how do we know if a cylinder does not corresponds to an object, i.e., how to deal with different number of detections for different scenes?
>
> ## **Response:**
> We thank the reviewer for this question. Our formulation can be intuitively understood in a way analogous to anchor-based detectors, where a fixed set of candidates is generated and then filtered based on confidence. In our case, each query acts as a 3D anchor (i.e., a candidate cylinder) that predicts both geometric parameters and an objectness score.
>
> To ensure sufficient coverage or adaptation to different scenes, the number of queries (anchors) is chosen to exceed the maximum expected number of objects in a scene. Each query predicts a classification score indicating whether it corresponds to a valid object or background ($\emptyset$). At inference time, we simply retain queries with high objectness scores and discard the rest. This design naturally produces a variable number of detections across different scenes, analogous to how anchor-based detectors filter low-confidence anchors, while ensuring that all true objects can be captured by the over-complete set of candidate cylinders.
>
> ## **Requested Change #4:**
> Currently Fig. 7 is not clear enough, as the projected points on 2D may overlap. It is hard for the readers to distinguish predictions and ground truths. A better way is to provide the qualitive comparison with the baselines in ImV as Fig. 5 and Fig. 6, and highlight the difference.
>
> ## **Response:**
> We thank the reviewer for this suggestion. We would like to respectfully note that the difficulty in distinguishing predictions from ground truths in Figure 7 is itself an indicator of high localization accuracy and precision. The visualization convention in Figure 7 is as follows: we only annotate regions where errors occur. Specifically, false positives (green circle) are marked where predictions have no corresponding ground truth, false negatives (black circle) where ground truths have no matching prediction, and low-precision (red circle) detections where predictions visibly deviate from ground truth. Unmarked regions indicate that predictions closely align with their ground truths and are considered correct. Since MVDGC achieves the highest MODP across benchmarks, the majority of its predictions tightly overlap with the ground truth, leaving fewer annotated errors and making the two markers hard to separate visually. In contrast, the baseline methods exhibit clearly visible errors because their predictions deviate from the ground truth.
>
> We agree that the visualization can be improved for better readability. We will enhance Figure 7 by using more distinct marker styles and adding zoomed-in insets of representative regions so that readers can more easily appreciate both the precision of our predictions and the errors of baseline methods. Following the reviewer's suggestion, we will also include a qualitative comparison with baselines in the image-view space in the Appendix. Since each scene involves 6–7 camera views across 4 methods, presenting this comparison in the main paper would require significant space. The Appendix provides a more suitable location for this detailed visualization.
>
> ## **Requested Change #5:**
> In the Tab.1 the right-most column, the horizontal lines for each cell are missing.
>
> ## **Response:**
> We thank the reviewer for pointing out this issue. We will increase the line thickness and row space to make the horizontal lines more apparent.

---

> ### Author Response · Authors · 2026-05-07
> **Follow up message**
>
> Dear Reviewer **PaA7**,
>
> We are kindly following up regarding our revision. We hope that our response has addressed your concerns well.
>
> We would greatly appreciate it if you could share with us any further concerns you may have.
>
> Thank you,
>
> Authors.

---

### Review · Reviewer_twWV · 2026-03-13

**Summary Of Contributions:**

This paper presents three main contributions. First, it introduces a query-based MVPD framework that represents each pedestrian as a sparse 3D cylindrical query (x, y, h, r), projects this query onto each view, and directly samples from raw ImV features. Second, it incorporates query relationships within and across views, along with object-specific adaptive view fusion, through three modules: Multi-view Feature Sampling, Intra-Inter Query Interaction, and Multi-view Adaptive Fusion. Third, we introduce dual geometric constraints that impose both BEV ground point regression and ImV bounding box supervision on the same cylindrical representation, handling not only detection but also ID propagation from BEV tracking to ImV tracking.

Key strengths include a clear problem formulation with explicit motivation to avoid dense projections onto BEVs. Results are exceptionally strong on WildTrack and MultiViewX datasets, reporting MODP 83.8 / MODA 94.2 / F1 97.1 on WildTrack and MODP 93.0 / MODA 95.7 / F1 97.8 on MultiViewX. Furthermore, the ablation studies in Tables 8–14 are commendable, as they provide a detailed examination of the contributions from each module, dual supervision, KL loss, Adaptive Hungarian matching, K parameter settings, query shapes, and query counts.

On the other hand, the weaknesses are as follows. First, while MODP 79.6 is the highest for GMVD, MODA 71.2, Recall 76.9, and F1 84.1 fall below MVFP. This does not align with the claim in the abstract and contributions section that “GMVD achieves SOTA across multiple evaluation metrics.” Second, the ImV tracking comparison pits a multi-view pipeline, which creates tracklets in BEV and propagates global IDs to 2D boxes, against an existing single-view tracker. While interesting for downstream applications, this is not a fair comparison against pure monocular tracking methods.

**Additional Comments:**

N/A

**Audience:**

Yes

**Audience Explanation:**

This paper spans multiple key topics: multi-view detection, object-centric query formulation, geometry representation design under weak 3D supervision, and BEV-ImV joint supervision. It should be of considerable interest to TMLR readers interested in multi-view vision, geometry-aware detection, and tracking.

**Claims And Evidence:**

No

**Claims Explanation:**

The core technical claim, namely, "linking BEV and ImV using cylindrical queries to achieve high-precision MVPD with WildTrack/MultiViewX" is quite well supported by experiments and ablation studies. Indeed, Table 9 shows that ImV-only performs stronger than BEV-only, and the best results are achieved when both are combined, demonstrating the effectiveness of dual geometric constraints.

However, when considering the paper's overall headline claim, the evidence supporting assertions like those regarding GMVD is not sufficiently consistent.

**Requested Changes:**

- Please revise the claims regarding GMVD. While the Abstract and Contribution sections state that GMVD achieves SOTA across multiple metrics, Table 7 shows that only MODP is the best, with MODA/Recall/F1 falling below MVFP. Therefore, regarding GMVD, either tone down the claim to "highest MODP" or “competitive generalization performance,” or provide additional evidence.
- Please specify the ablation settings. For Table 8–14, it is unclear from the table itself which dataset/split was used. While Table 7 for GMVD is explicitly stated, the benchmark for the remaining ablations is ambiguous. Please add the dataset to each table.
- Please add a report on computational complexity and efficiency. The query count is 768, the decoder has 4 layers, and it includes multi-view interaction, though there are concerns about speed due to this. Specifying runtime, FPS, latency, memory usage, and parameter count would clarify its practical positioning for real-world deployment.

---

> ### Author Response · Authors · 2026-04-08
> **Rebuttal by Authors (1)**
>
> We sincerely thank the reviewer for the thorough and constructive review. We are grateful that the reviewer recognized the core contributions of our work, particularly the cylindrical query representation, the three-module design for multi-view interaction, and the dual geometric constraints enabling BEV-to-ImV tracking correspondence. We are also encouraged that the reviewer found the problem formulation clearly motivated and acknowledged the strong experimental results on WildTrack and MultiViewX, as well as the comprehensive ablation studies. We address each concern below.
>
> ## **Weakness #1 and Requested Change #1:**
> - **Weakness #1**: First, while MODP 79.6 is the highest for GMVD, MODA 71.2, Recall 76.9, and F1 84.1 fall below MVFP. This does not align with the claim in the abstract and contributions section that “GMVD achieves SOTA across multiple evaluation metrics.”
> - **Request Change #1**:  Please revise the claims regarding GMVD. While the Abstract and Contribution sections state that GMVD achieves SOTA across multiple metrics, Table 7 shows that only MODP is the best, with MODA/Recall/F1 falling below MVFP. Therefore, regarding GMVD, either tone down the claim to "highest MODP" or “competitive generalization performance,” or provide additional evidence.
>
> ## **Response:**
> We thank the reviewer for this observation. We would like to clarify that the original claim refers to achieving state-of-the-art on multiple metrics collectively across the three datasets, not on every metric of every dataset. The phrasing was intended to group the datasets together to highlight that MVDGC was evaluated on both standard MVPD benchmarks and a generalization benchmark. To address the concern raised by reviewer, we will revise both the Abstract and Contribution sections to be more precise.
>
> *"Extensive experiments and ablation studies demonstrate that MVDGC achieves state-of-the-art performance across multiple evaluation metrics on MVPD benchmarks, including WildTrack and MultiViewX. On the generalized multi-view detection (GMVD) dataset, MVDGC achieves the highest MODP and precision, while maintaining competitive performance on the remaining metrics, highlighting its robustness and generalization to unseen scene configurations."*
>
> ## **Weakness #2:**
> The ImV tracking comparison pits a multi-view pipeline, which creates tracklets in BEV and propagates global IDs to 2D boxes, against an existing single-view tracker. While interesting for downstream applications, this is not a fair comparison against pure monocular tracking methods.
>
> ## **Response:**
> We appreciate the reviewer's concern and agree that a direct performance comparison between a multi-view pipeline and a monocular tracker would be unfair. We would like to clarify that this is not the intention of the comparison. Specifically, the BEV-ImV tracking experiment serves as the validation for our proposed cylindrical representation and its functionality. We don't propose the novel tracking method in both domains. To support the validation, the monocular trackers in Tables 4 and 6 serve purely as quantitative reference points. Without them, readers would have no basis to assess whether the propagated ImV tracklets are functionally meaningful. We note that Tables 4 and 6 already include a "Global ID" column that explicitly distinguishes MVDGC (which uses global BEV identities) from monocular methods (which do not), making the difference in setup transparent. Moreover, the intention of BEV-ImV tracking is reflected in how we frame the experiment and present in Section 4.5.2 (Application of 3D Cylindrical Representation to BEV and ImV Pedestrian Tracking) which is separated from the main MVPD performance.
>
> In Section 4.5.2, we focus on two specific validation purposes:
> - **First, BEV and ImV tracking validates the geometric consistency of our representation.** To our knowledge, no prior MVPD method has established an explicit correspondence between BEV tracklets and ImV bounding boxes. The ImV tracking results demonstrate that identities assigned in BEV can be seamlessly propagated to image views through our cylindrical projection, a unique capability enabled by our unified cylindrical representation.
> - **Second, BEV tracking validates our detection quality.** We intentionally adopted a minimal off-the-shelf tracker (ByteTrack with Euclidean distance association in BEV space), with no appearance features, no re-identification module, and no temporal modeling. The strong tracking results with such a simple pipeline confirm that the performance gains originate from the detection quality and geometric consistency of our framework.
>
> The concluding sentence of Section 4.5.2 makes the intention explicit: *"These results highlight the robustness of tracklet transfer enabled by our approach, and verify the effectiveness of the proposed 3D cylindrical query as a strong BEV-ImV bridge for maintaining global identities across views."*

---

> ### Author Response · Authors · 2026-04-08
> **Rebuttal by Authors (2)**
>
> ## **Requested Change #2:**
> Please specify the ablation settings. For Table 8–14, it is unclear from the table itself which dataset/split was used. While Table 7 for GMVD is explicitly stated, the benchmark for the remaining ablations is ambiguous. Please add the dataset to each table.
>
> ## **Response:**
> We thank the reviewer for this suggestion. We will add the dataset specification to the caption of each ablation table (Tables 8–14) to make the experimental setting unambiguous. All ablation studies are conducted on the WildTrack dataset, a real-world benchmark in the MVPD topic. We will revise the captions accordingly.
>
> | Original Caption | Updated Caption |
> |-----------------|-----------------|
> | Table 9: Dual Geometric Constraints | Table 9: Dual Geometric Constraints on the WildTrack dataset |
> | Table 10: Ablation study on different types of losses | Table 10: Ablation study on different types of losses on the WildTrack dataset |
> | Table 11: Ablation study on various assignment strategies | Table 11: Ablation study on various assignment strategies on the WildTrack dataset |
> | Table 12: Ablation study on number of assignments | Table 12: Ablation study on number of assignments on the WildTrack dataset |
> | Table 13: Ablation study on various query types | Table 13: Ablation study on various query types on the WildTrack dataset |
> | Table 14: Ablation study on different number of queries | Table 14: Ablation study on different number of queries on the WildTrack dataset |
>
> ## **Requested Change #3:**
> Please add a report on computational complexity and efficiency. The query count is 768, the decoder has 4 layers, and it includes multi-view interaction, though there are concerns about speed due to this. Specifying runtime, FPS, latency, memory usage, and parameter count would clarify its practical positioning for real-world deployment.
>
> ## **Response:**
> We thank the reviewer for this suggestion. We report the computational complexity of MVDGC below, benchmarked on a single NVIDIA RTX A6000 with input resolution $640 \times 640$:
>
> | Params (M) | FPS | GPU Memory (GB) |
> |-----------|-----|----------------|
> | 64 | 3.8 | 32.71 |
>
> We note that the relatively high query count (768) is necessary due to the nature of the pedestrian detection task. For instance, WildTrack spans $12 \times 36 m$ (432$m^2$) and MultiViewX covers $16 \times 25 m$ (400$m^2$), both monitored and labeled at 2.5 cm grid resolution. In addition, pedestrians are significantly smaller in scale and appear in much denser configurations, requiring sufficient query density to adequately cover the ground plane. In terms of real-world deployment, all three of the MVPD datasets (WildTrack, MultiViewX, and GMVD) are annotated at 2 FPS, which is the standard operational frame rate. With the achieved FPS of 3.8, MVDGC comfortably operates in real-time for its target surveillance application.

---

> ### Author Response · Authors · 2026-05-07
> **Follow up message**
>
> Dear Reviewer **twWV**,
>
> We are kindly following up regarding our revision. We hope that our response has addressed your concerns well.
>
> We would greatly appreciate it if you could share with us any further concerns you may have.
>
> Thank you,
>
> Authors.

---

### Review · Reviewer_m7Zm · 2026-03-28

**Summary Of Contributions:**

This is a work belonging to the 3d detection and tracking area broadly from the DETR setup. The main idea is to adapt queries to cylindrical queries so that it projects to a 2D box in the image plane. Then they supervise in multiview fashion - take boxes in each view and supervise with GT 2d boxes. For 3D supervision, they seem to supervise box points - no yaw, size - and seem to do away with heatmap type estimations in 3D. As regards motivation, it is argued that the projection in standard methods (e.g. BEVFormer) leads to distortion, and these methods aren't using 2D supervision, thus not handling occlusions (especially important for pedestrians). They show tracking and detection results which show efficacy of the method. Localization metrics are favoured.

Metics are good, but tracking is overemphasized - I assume that this is to show that the queries can more effectively capture identity information. Even so, the tracking method is not novel, nor does it apply any fittings to ByteTrack (now multiview).

My impressions:
Pros
-----
+ Clear motivation, showing occlusion can be handled through multiview supervision
+ Clever design of cylindrical query to get around irregular polygon projection
+ Multiview supervision clearly designed
+ Good, convincing evaluation with main focus on what seems to be localization metrics

Cons
------
- Tracking overemphasized, not novel
- No evaluation on nuscenes or waymo
- Reliance on 2D GT (this is not so much a con in itself, maybe even a pro but its curious)
- Methods compared seem to be a bit old and slightly unfair. What about BEVFormer and PETR, etc.
- There are some qualitative visuals, but I would love to see a detailed walkthrough of failures in existing methods versus theirs.

Overall, I like this work and would recommend publication. Occlusion is a big issue in these kinds of setups. Many existing methods tend to be gloss over this issue. Multiview supervision goes a long way in handling them. The paper does seem though to be a little rough around the edges, and would benefit from some polishing with examples (clearly described) showing how their method overcomes shortcomings in other methods.

**Additional Comments:**

Great work, and very interesting adaptations to the network. I would like to see some of the improvements as suggested above.

**Audience:**

Yes

**Audience Explanation:**

This topic is of core interest to the computer vision and autonomous driving community and general scene understanding.

**Claims And Evidence:**

Yes

**Claims Explanation:**

Qualitative and quantitative evaluation show effectiveness of the method.

**Requested Changes:**

Please polish the tracking emphasis, and maybe describe the approach in more detail. I am quite familiar with ByteTrack and its emphasis on occlusions, but it would help to expand the description, especially because the paper emphasizes tracking treatment.

Please also compare with query based methods (BEVFormer, PETR, StreamPETR, etc.)

---

> ### Author Response · Authors · 2026-04-08
> **Rebuttal by Authors (1)**
>
> We sincerely thank the reviewer for the thorough and insightful review. We are encouraged by the recognition of our core contributions, including the clear motivation for handling occlusion through multi-view supervision, the design of the cylindrical query to avoid irregular polygon projections, the well-designed multi-view supervision scheme, and the convincing evaluation with strong localization results. We are also grateful that the reviewer finds this work interesting and recommends publication. We address each concern below.
>
> ## **Con #1 and Requested Change #1:**
>
> - Con #1: Tracking overemphasized, not novel.
>
> - Requested Change #1: Please polish the tracking emphasis, and maybe describe the approach in more detail. I am quite familiar with ByteTrack and its emphasis on occlusions, but it would help to expand the description, especially because the paper emphasizes tracking treatment.
>
> ## **Response:**
> We thank the reviewer for this feedback. Regarding tracking emphasis and description along with ByteTrack, we would like to clarify that our main contribution is the proposed 3D cylindrical query formulation for multi-view pedestrian detection, together with the dual BEV–ImV geometric constraints. Our intention is not to claim a new tracking algorithm. Rather, the tracking experiments are included as secondary validation of the proposed representation and its geometric properties.
>
> More specifically, the tracking experiments serve two purposes.
> - **First, BEV-ImV tracking validates the geometric consistency of our representation.** A key novelty of MVDGC is its 3D cylindrical representation, which creates a natural bridge between BEV and ImV domains. Each cylinder encodes a BEV ground point and produces 2D bounding boxes across all visible camera views through geometric projection. When a cylinder is assigned an identity during BEV tracking, that identity automatically and consistently transfers to every camera view. The BeV and ImV tracking results in Tables 3, 4, 5, 6 validate that the learned representation preserves cross-view geometric correspondence and globally consistent identities. Prior MVPD methods simply cannot perform this task because they lack the geometric correspondence between BEV identities and ImV detections. The large margins over dedicated single-view trackers (e.g., +42.1 MOTA over MeMOTR on WildTrack) demonstrate that globally consistent identities from BEV space provide a powerful advantage over methods that reason about each camera independently.
> - **Second, BEV tracking further validates our detection quality.** In tracking-by-detection, tracking performance is largely determined by the quality of the underlying detections. To isolate this factor, we intentionally adopted an off-the-shelf tracker (ByteTrack with Euclidean distance association in BEV space), with no appearance features, no re-identification module, and no temporal modeling. Our strong BEV tracking performance in Tables 3 and 5 is largely driven by the superior detection quality shown in Table 2. Despite using only simple tracker, MVDGC outperforms EarlyBird, which incorporates a dedicated re-identification branch, and TrackTacular, which explicitly exploits temporal information from previous frames. This confirms that the performance gains originate from the detection quality of our framework rather than from a sophisticated tracking algorithm.

---

> ### Author Response · Authors · 2026-04-08
> **Rebuttal by Authors (2)**
>
> ## **Cons #2, #4 and Requested Change #2 :**
>
> - Con #2: No evaluation on nuscenes or waymo.
>
> - Con #4: Methods compared seem to be a bit old and slightly unfair. What about BEVFormer and PETR, etc.
>
> -  Requested Change #2: Please also compare with query based methods (BEVFormer, PETR, StreamPETR, etc.)
>
> ## **Response:**
> We sincerely thank the reviewer for the thoughtful and constructive feedback. We would like to respectfully clarify a distinction that may help contextualize our benchmark and baseline choices. Our work addresses **multi-view pedestrian detection (MVPD)**, which operates in a fundamentally different setting from **3D object detection in autonomous driving**. Although both topics involve multi-camera input and BEV reasoning, they differ substantially in camera configuration, available annotations, and task objectives. We highlighted this distinction in Table 1 of our manuscript, where the *3D Box Query* column (the first column) corresponds to autonomous driving methods (e.g., DETR3D, BEVFormer), while the remaining columns (anchor-based, centralized-based, and our proposed cylindrical query) correspond to the MVPD setting. To further clarify, we summarize the key differences between the two problem settings in the table below:
>
> |  | **MVPD (Ours)** | **3D Object Detection (Autonomous Driving)** |
> |--|-----------------|--------------------|
> | **Camera Setup** | Fixed, static cameras pointing **inward** with high overlap | Moving ego-centric camera rig pointing **outward** with low overlap |
> | **Annotations** | Ground-plane locations (x, y) and 2D bounding boxes per view | Full 3D bounding boxes (x, y, z, w, h, l, roll, pitch, yaw) |
> | **Task Objective** | Localize pedestrians on a shared ground plane | Estimate full 3D bounding box |
> | **Benchmarks** | WildTrack, MultiViewX, GMVD | nuScenes, Waymo, Argoverse |
> | **Representative Methods** | MVDet, MVDetr, 3DROM, EarlyBird, MVFP, PVH | BEVFormer, PETR, StreamPETR, DETR3D |
>
> As stated in the above table, methods such as BEVFormer, PETR, and StreamPETR belong to the autonomous driving 3D object detection setting, which differs fundamentally from MVPD. These methods require full 3D bounding box supervision that is unavailable in MVPD benchmarks, making direct comparison infeasible without substantial architectural modification.
>
> Our baselines (MVDet, MVDetr, 3DROM, EarlyBird, MVFP, MVTT, TrackTacular, PVH, among others) represent the current state-of-the-art specifically within the MVPD topic, and we have benchmarked on WildTrack, MultiViewX, and GMVD, which are the established evaluation protocols for this task. We believe that our comparisons are comprehensive and fair within this problem domain. Furthermore, we respectfully note that our compared methods are not outdated. Our baselines include MVFP (WACV 2024), EarlyBird (WACV 2024), TrackTacular (CVPR 2024), and PVH-Enhanced (CVPR 2025), all published within the last one to two years.
>
> ## **Con #3:**
> Reliance on 2D GT (this is not so much a con in itself, maybe even a pro but its curious)
>
> ## **Response:**
> We appreciate the reviewer raising this point. We would like to offer some clarification on how 2D bounding box annotations are used in our framework and in the broader MVPD literature. Our method does not rely on 2D ground-truth boxes at inference time; they are used only as training supervision. Moreover, per-view 2D bounding box annotations are a standard component of MVPD datasets alongside BEV ground plane locations. These annotations are not an additional requirement introduced by our method; they are part of the existing dataset that all methods in this topic have access to.
>
> The key difference lies in how this supervision is used. In existing MVPD methods, 2D bounding boxes are either leveraged to refine foreground object features or independently detected in each camera view and subsequently associated across views using geometric and appearance cues. As a result, the ImV and BEV objectives remain decoupled, and the 2D supervision does not directly constrain the final BEV localization.
>
> In contrast, MVDGC establishes a direct geometric correspondence between BEV ground points and image-view bounding boxes through the 3D cylindrical representation. The cylinder's center defines the BEV location, and its projection through camera parameters produces the 2D bounding box, whose outputs are derived from a single unified representation. Consequently, the ImV supervision is not merely improve feature quality; it directly regularizes the shared 3D representation and helps refine the BEV localization by enforcing cross-view geometric consistency. As demonstrated in Table 9, joint BEV-ImV supervision yields substantial gains over either alone (e.g., +8.5 MODA and +10.4 MODP compared to BEV-only).
>
> Therefore, we view the utilization of 2D ground-truth boxes as a strength of our formulation as it enables the model to more fully exploit annotations that already exist in MVPD datasets.

---

> ### Author Response · Authors · 2026-04-08
> **Rebuttal by Authors (3)**
>
> ## **Con #5:**
> There are some qualitative visuals, but I would love to see a detailed walkthrough of failures in existing methods versus theirs.
>
> ## **Response:**
> We thank the reviewer for this suggestion. Fig 7 in our manuscript provides a qualitative comparison across methods, but we agree that a more detailed narrative walkthrough would strengthen the paper. We provide the analysis below and will incorporate it into the revised manuscript.
>
> We observe two primary failure regions across existing methods: *crowded central areas* and *scene borders*.
> - In **crowded central areas**, pedestrians frequently occlude each other across multiple views, causing centralized-based methods (MVDeTr, 3DROM, EarlyBird) to produce imprecise and misaligned predictions. For example, on the WildTrack dataset, MVDeTr struggles to precisely locate objects in the middle of the scene. A fundamental reason is that these methods rely heavily on foot-point features projected onto the BEV plane. In crowded scenes, the feet of nearby pedestrians overlap and occlude each other, producing ambiguous and entangled features at the ground level. This problem is further exacerbated by perspective projection, which compresses overlapping pedestrians into similar BEV locations on the dense heatmap, making it even more difficult to distinguish individual ground points. MVDGC avoids this limitation by sampling features from the full body extent of each pedestrian through the cylindrical projection. When foot regions are occluded, features from the torso, shoulders, or head still contribute to the localization, providing complementary cues that help resolve the pedestrian's ground-plane position even under heavy lower-body occlusion.
> - At **scene borders**, objects may only be clearly visible from one or two cameras, or appear too small in distant views to be reliably detected. 3DROM exhibits several low-precision predictions near the border. Although its random occlusion augmentation strategy improves robustness in central regions,  3DROM does not address the low precision at the periphery. EarlyBird shows both false negatives and false positives near scene borders, where insufficient multi-view overlap leads to ambiguous feature aggregation on the BEV plane.
>
> MVDGC mitigates both failure modes through its object-centric design. For crowded regions, the cylindrical queries avoid the BEV feature aggregation problem. Each query independently regresses its position rather than competing for activation on a shared dense map. For border regions, the Multi-view Adaptive Fusion module (Section 3.5) learns instance-specific attention weights that can prioritize the one or two views where the pedestrian is clearly visible, rather than uniformly averaging features from all cameras including those with poor visibility.
>
> We will revise Section 4.6 and add this detailed error walkthrough accordingly.

---

> ### Author Response · Authors · 2026-05-07
> **Follow up message**
>
> Dear Reviewer **m7Zm**,
>
> We are kindly following up regarding our revision. We hope that our response has addressed your concerns well.
>
> We would greatly appreciate it if you could share with us any further concerns you may have.
>
> Thank you,
>
> Authors.

---

### Decision · Action_Editor_6ka3 · 2026-05-28

**Recommendation:** Accept as is

**Additional Comments:**

No revision is requested. No certification is recommended after comparing the quality of this submission with the related criteria.

**Audience:**

Yes

**Audience Explanation:**

This topic is of significant interest to the computer vision, autonomous driving, and general scene understanding communities. It should also attract researchers working in related areas such as multi-view vision, geometry-aware object detection, and pedestrian detection and tracking.

**Claims And Evidence:**

Yes

**Claims Explanation:**

Two reviewers commented that the qualitative and quantitative evaluations demonstrate the effectiveness of the proposed method, and that the claims are well supported by accurate, convincing, and clear evidence. The third reviewer noted that the evidence supporting certain assertions, particularly those regarding GMVD, was not sufficiently consistent. The authors provided an extensive and clear rebuttal. Following the rebuttal, all three reviewers expressed positive opinions on the Claims and Evidence criterion and recommended either Accept or Leaning Accept.

---

> ### Author Response · Authors · 2026-05-29
> **Thank you**
>
> We sincerely thank the Editor-in-Chief, Action Editors, and three reviewers for recognizing the contribution of our work. We are truly grateful for the positive decision and for accepting our manuscript as is.